# Tunable and nonlinearity-enhanced dispersive-plus-dissipative coupling in photon-pressure circuits

Mohamad Kazouini ✉, Janis Peter, Zisu Emily Guo, Benedikt Wilde, Kevin Uhl, Dieter Koelle, Reinhold Kleiner & Daniel Bothner ✉

Photon-pressure circuits are the circuit implementation of the cavity optomechanical Hamiltonian and discussed for qubit readout, low-frequency quantum photonics and dark matter axion detection. Due to the enormous design flexibility of superconducting circuits, photon-pressure systems provide fascinating possibilities to explore unusual parameter regimes of the optomechanical Hamiltonian. Here, we report the realization of a photon-pressure platform, in which a GHz circuit interacts with a MHz circuit via a magnetic-flux-tunable combination of dispersive and dissipative photon-pressure. In addition, both coupling rates are considerably enhanced by nonlinearities of the GHz-mode, which leads to the multi-photon coupling rates scaling stronger with the pump photon number $n_c$ than the usual $\sqrt{n_c}$ dependence. We demonstrate that interference of the two interaction paths leads to a Fano-like response in photon-pressure induced transparency, and that the dynamical backaction is considerably modified compared to the dispersive case, including a parametric instability caused by a red-detuned pump tone.

Superconducting microwave circuits have developed into one of the most exciting and versatile platforms for emerging technologies in both the quantum and the classical domain[1,2]. Quantum processors[3,4], quantum simulators[5,6], frequency transducers[7,8], electron-spin-resonance detectors[9,10], or quantum-limited signal amplifiers[11-13] are just a few of the many groundbreaking results of the recent decade. Being highly flexible in frequency, linewidth and nonlinearity, it seems one can engineer a superconducting circuit implementation for any imaginable application on demand. One of the more recent developments in superconducting microwave platforms are photon-pressure circuits, i.e., the circuit quantum electrodynamics implementation of the optomechanical Hamiltonian, in which two oscillators are parametrically interacting through a radiation-pressure-like nonlinear coupling[14-16]. Compared to cavity optomechanics[17], photon-pressure circuits provide a large degree of designability and possess orders of magnitude larger single-photon coupling rates[18,19], which makes them

not only interesting for applications, but also to study new regimes of radiation-pressure physics. Beyond a high-quality replication of optomechanical gold-standard experiments such as photon-pressure induced transparency[15,16], parametric strong coupling[16] or sideband-cooling into the quantum regime[18], photon-pressure circuits have allowed the realization of more sophisticated experiments with parametrically amplified coupling rates, nonreciprocal bath dynamics or effective negative mass modes based on their intrinsic Kerr nonlinearity[20,21].

All the existing experiments to date are based on dispersive photon-pressure, which is realized when the amplitude of one LC oscillator modulates the resonance frequency of a second. However, this is not the only possible interaction type, the amplitude of one oscillator might also modulate the decay rate of the other. The latter then corresponds to so-called dissipative photon-pressure, a type of radiation-pressure that has attracted considerable attention in

---

Physikalisches Institut, Center for Quantum Science (CQ) and LISA+, Universität Tübingen, 72076 Tübingen, Germany. ✉e-mail: mohamad.adnan-el-kazouini@uni-tuebingen.de; daniel.bothner@uni-tuebingen.de

optomechanical systems[22–34], especially from the theoretical viewpoint[22–30] since it is not as straightforward to realize experimentally as its dispersive counterpart[31–34]. Dissipative coupling would for instance allow groundstate-cooling in the sideband-unresolved limit[22], i.e., to extend the quantum regime to lower photon frequencies, or to enable alternative protocols for squeezing and quantum-limited sensing[25–27]. Therefore, the implementation of dissipative coupling would greatly enrich the capabilities of photon-pressure circuits, constitute a flexible testbed for dissipative optomechanics and enable kHz to MHz quantum photonics as e.g. required for dark matter axion detection with radio-frequency upconverters[35–37].

Here, we report the implementation of niobium-based photon-pressure circuits with both a dispersive and a dissipative interaction. The two participating LC circuits are coupled to each other through an asymmetrically shared nano-constriction-based superconducting quantum interference device (SQUID). The dissipative single-photon coupling rate of the system is flux-tunable with large values up to $g_{0\kappa}/2\pi \sim 150$ kHz, and—in contrast to earlier experiments with dissipative radiation-pressure in optomechanical systems—it is solely attributed to a modulation of the internal decay rate. As a consequence of the interference between the dispersive and the dissipative interactions, we observe a Fano-like modification of photon-pressure induced transparency, that provides an independent and simple measure to quantify the ratio between the two coupling rates. Since the GHz SQUID circuit possesses both a flux-dependent Kerr nonlinearity and

nonlinear damping, we furthermore observe that the multi-photon coupling rates are not scaling $\propto \sqrt{n_c}$ with the pump photon number $n_c$, but have contributions $\propto \sqrt{n_c^3}$ and $\propto \sqrt{n_c^7}$. Finally, we characterize the generalized dynamical backaction of the high-frequency (HF) intracavity fields to the low-frequency (LF) circuit in the presence of a red-sideband pump field. We find characteristic signatures of dissipative coupling contributions as well as a pump-power scaling in agreement with the nonlinearity-enhanced cooperativities. Our results lay the foundation for the experimental investigation of dissipative and nonlinearity-enhanced interactions in photon-pressure systems and related platforms like SQUID optomechanics[38–45], and constitute a stepping stone towards photon-photon backaction-cooling in the unresolved sideband regime and kHz quantum photonics. Facilitated by the use of niobium, all experiments presented here could be conducted at liquid helium temperatures, also opening the door for investigating radiation-pressure physics in the thermal regime and for photon-pressure experiments without the need for dilution refrigerators.

## Results

### Device and setup

Our photon-pressure device, discussed in Fig. 1 and Supplementary Notes 2 and 3, comprises a low-frequency LC circuit with a sweetspot resonance frequency $\Omega_0 = 2\pi \times 446.7$ MHz and a high-frequency quantum interference circuit with a sweetspot resonance frequency

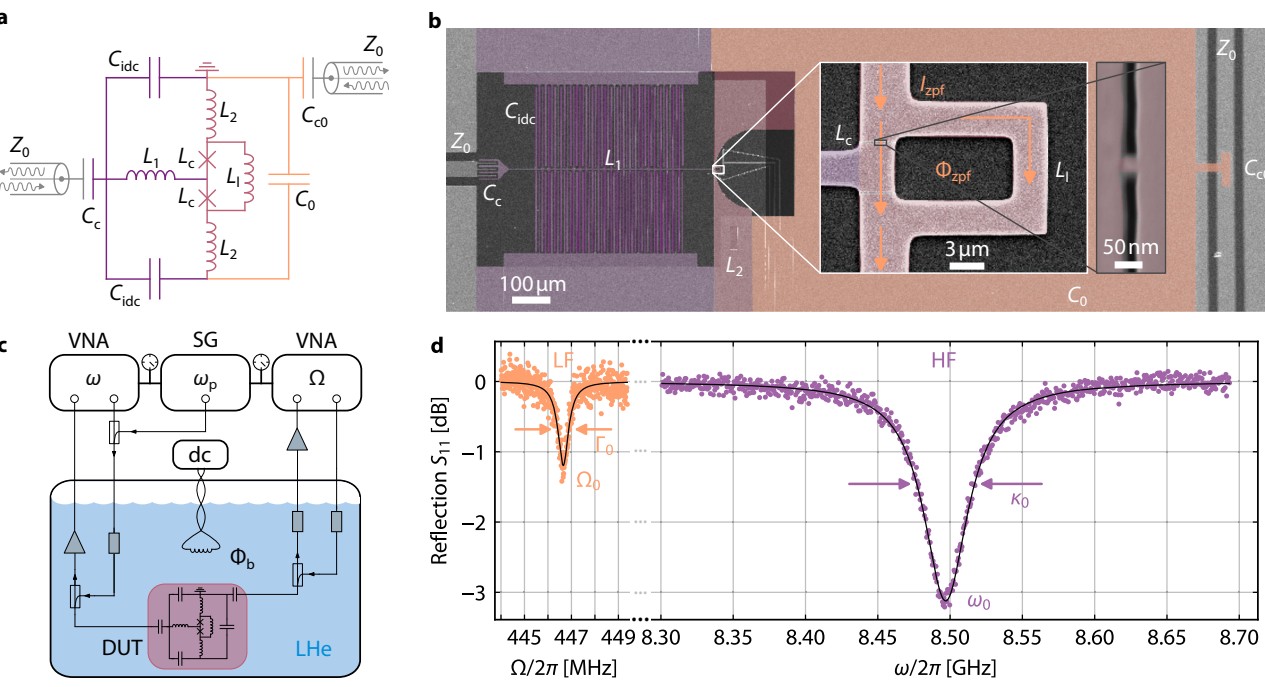

**Fig. 1 | Niobium photon-pressure circuits based on a monolithic nano-constriction SQUID and operated in liquid helium. a** Simplified circuit equivalent. The HF mode (left half, purple and pink) combines two interdigitated capacitors $C_{idc}$ with the inductors $L_1$, $L_2$ and $L_c$. The LF mode (right half, orange and pink) comprises a parallel-plate capacitor $C_0$ with an inductance composed of $L_2$, $L_1$ and $L_c$. Both modes share the SQUID and parts of the linear inductance (in pink), and each mode is coupled to an individual feedline by coupling capacitors $C_c$ and $C_{c0}$, respectively. **b** False-color scanning electron microscopy image of the device. Colored and light gray parts are Nb, dark gray and transparent parts are Si substrate and $Si_3N_4$ dielectric of the LF capacitor, respectively. Color-code equivalent to **a**. White box inset shows a magnification of the interaction part. The rectangle is the SQUID loop, and the LF zero-point-fluctuation current $I_{zpf}$ flows through the SQUID from top to bottom, hereby threading the loop with a zero-point-fluctuation flux $\Phi_{zpf}$. Black box inset shows one of the monolithic 3D nano-constrictions with inductance $L_c$ (image

rotated by 90°). **c** Simplified experimental setup. The device under test (DUT) is immersed in liquid helium (LHe) and connected to two coaxial lines for readout of the individual modes by a vector network analyzer (VNA). A HF sideband-pump field with frequency $\omega_p$ from a signal generator (SG) is combined with the HF VNA signal in a directional coupler. Input and output of each readout line are combined by means of further directional couplers and the HF return signal is amplified by a cryogenic amplifier depicted as triangle; the LF mode return signal is amplified by a room-temperature amplifier. A small magnetic-field coil is driven with a direct current (dc) source to induce a bias flux $\Phi_b$ into the SQUID. Gray rectangles represent attenuators. For more details cf. Supplementary Note 1. **d** Reflection $S_{11}$ from both circuits ($\Phi_b = 0$) measured through their individual feedlines. Colored symbols are experimental data, black lines are fit curves. From the fits, we extract the resonance frequencies $\omega_0 = 2\pi \times 8.497$ GHz and $\Omega_0 = 2\pi \times 446.7$ MHz, as well as the total decay rates $\kappa_0 = 2\pi \times 44.9$ MHz and $\Gamma_0 = 2\pi \times 525$ kHz.

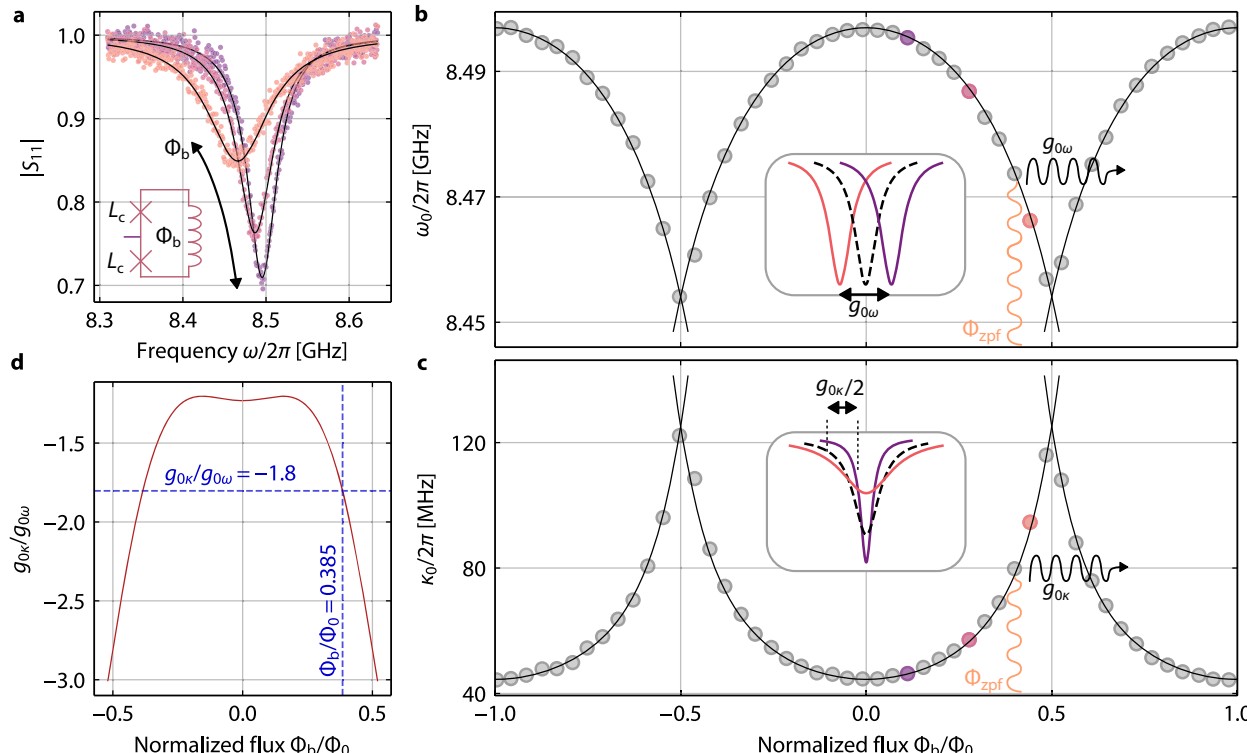

**Fig. 2 | Dispersive and dissipative photon-pressure coupling rates and their tuning with bias flux through the SQUID.** **a** Reflection $|S_{11}|$ of the HF mode for three different bias-flux values $0.1 \lesssim \Phi_b/\Phi_0 \lesssim 0.45$. With increasing bias flux, the resonance frequency shifts to lower values and the total linewidth increases. Symbols are data, lines are fits. The frequency-shift can be understood as an increase of the nonlinear constriction inductance $L_c$ with increasing flux through the SQUID loop (cf. inset), the increase in linewidth by current-induced quasiparticles. **b** Resonance frequencies $\omega_0(\Phi_b)$, extracted from fits to reflection data $S_{11}$ (cf. **a**), which show a periodic modulation with period $\Phi_0$. Circles are data, line is a fit (cf. Supplementary Note 3). If the SQUID is biased to a constant value $\Phi_b/\Phi_0 \approx 0.385$ and additional oscillating zero-point-fluctuation flux from the LF mode $\Phi_{zpf} \approx 198.3\,\mu\Phi_0$ threads the SQUID, this leads to a modulation of the HF

resonance frequency. The total frequency-modulation induced by $\Phi_{zpf}$ is equivalent to the dispersive single-photon coupling rate $g_{0\omega} = 2\pi \times 27.1$ kHz. Inset schematic visualizes $g_{0\omega}$. **c** Analogous to panel **b**, but for the total HF linewidth $\kappa_0$, which periodically modulates between ⁓ 44.5 MHz and ⁓ 125 MHz, revealing that despite the large decay rates the device is in the sideband-resolved limit with $\Omega_0/\kappa_0 \gtrsim 3.5$ for all $\Phi_b$. Circles are data, line is a polynomial fit (cf. Supplementary Note 3). At the operation point $\Phi_b/\Phi_0 = 0.385$, the LF flux $\Phi_{zpf}$ induces a dissipative photon-pressure interaction with single-photon coupling rate $g_{0\kappa} = -2\pi \times 48.8$ kHz. Inset schematic visualizes $g_{0\kappa}$. **d** Ratio of dissipative to dispersive single-photon coupling rate $g_{0\kappa}/g_{0\omega}$ vs. $\Phi_b$, which is calculated from the fit curves in **b** and **c**. At the operation point for this work (marked by the crossing point of the blue dashed lines), we find $g_{0\kappa}/g_{0\omega} \approx -1.8$.

$\omega_0 = 2\pi \times 8.497$ GHz. Both circuits are composed of linear capacitors and multiple linear inductors (cf. Supplementary Note 2) and share as nonlinear inductance and central coupling element a rectangular SQUID. The Josephson elements in the SQUID are monolithic 3D nano-constriction junctions, fabricated by focused neon-ion-beam milling[46,47]. For maximized photon-pressure coupling rates, the SQUID is galvanically shared by the two circuits, and the LF zero-point-fluctuation currents induce a zero-point-fluctuation flux $\Phi_{zpf} \approx 198.3\,\mu\Phi_0$ in the SQUID loop (containing both magnetic and kinetic contributions) with the flux quantum $\Phi_0 \approx 2.068 \times 10^{-15}$ T m², cf. Supplementary Note 3. Each of the two circuits is capacitively coupled to an individual coplanar waveguide (CPW) feedline with characteristic impedance $Z_0 \approx 50\,\Omega$ for driving and readout. Due to the galvanic coupling, the circuits can also be considered a single circuit with a LF and a HF mode, and we will use the words circuit and mode interchangeably in this manuscript.

As superconducting material for all the metallic parts we chose dc-magnetron sputtered niobium with a critical temperature $T_c \approx 9$ K, and as substrate we use high-resistivity intrinsic silicon with a substrate thickness $t_{Si} = 525\,\mu$m. The bottom niobium layer – defining the HF circuit, the HF CPW feedline center conductor, the SQUID and the bottom electrode of the LF circuit capacitance – has a thickness $t_{Nb1} = 120$ nm and the top layer – defining the top electrode of the LF

capacitance, the LF CPW feedline center conductor, and all ground planes – has $t_{Nb2} = 300$ nm. As dielectric for the LF parallel-plate capacitors $C_0$ and $C_{c0}$ we deposited 200 nm of silicon-nitride by means of plasma-enhanced chemical vapour deposition. All layers have been patterned by maskless optical lithography and reactive-ion-etching (bottom Nb layer) or liftoff ($Si_3N_4$ and top Nb layer). Further details regarding device fabrication can be found in "Methods–Device fabrication".

For the experiments, the $10 \times 10$ mm²-large chip is wire-bonded to a microwave printed circuit board (PCB) and enclosed in a radiation-tight copper housing. A small electromagnetic coil is mechanically attached to the copper housing in order to apply a dc magnetic field perpendicular to the chip surface and to flux-bias the SQUID. For microwave control and response measurements, both on-chip fee-dlines are connected via CPWs on the PCB and PCB-SMP connectors to two individual coaxial cables, which combine both input and output (I/O) for each of the circuits by means of a directional coupler. The HF coaxial input line is highly attenuated to equilibrate the feedline thermal noise to the experiment temperature $T_s = 4.2$ K, and the HF output line is equipped with a cryogenic high-electron-mobility transistor (HEMT) amplifier to maximize the signal-to-noise ratio. On the LF side, both input and output line are slightly attenuated and the returning signal is amplified by a room-temperature HEMT. The

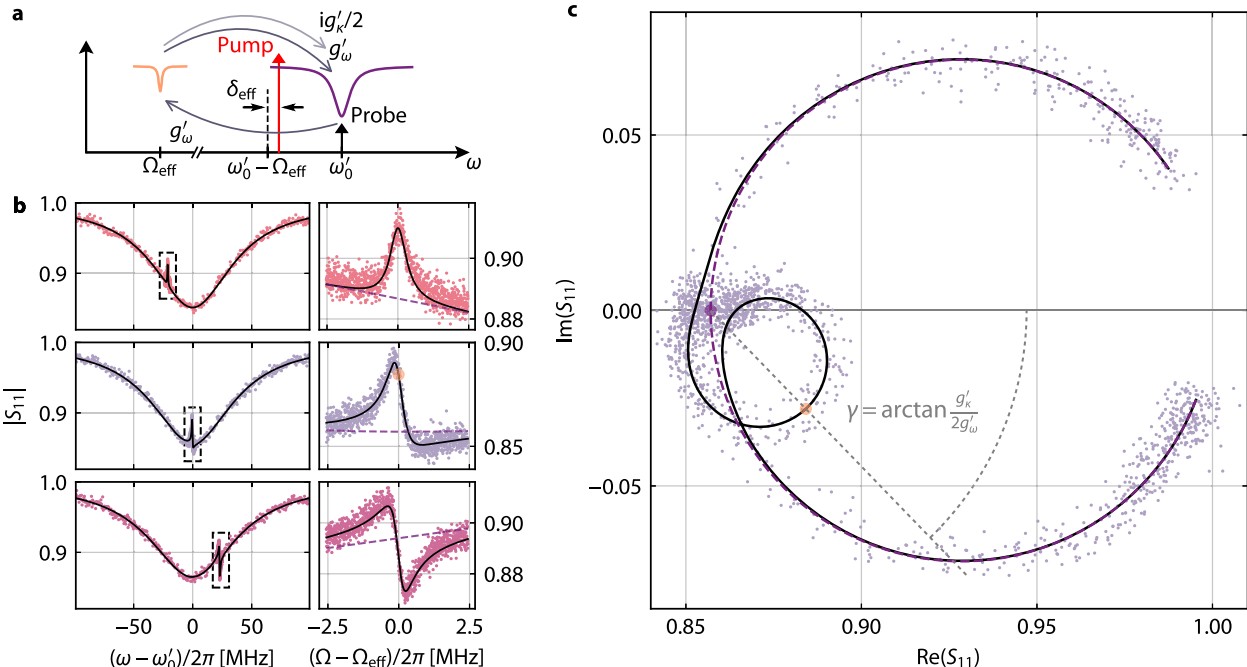

**Fig. 3 | Fano-transparency indicates interference of dispersive and dissipative photon-pressure. a** Schematic of the experimental setting for the observation of photon-pressure induced transparency (PPIT). A strong microwave pump tone (red vertical arrow) with power $P_p$ and frequency $\omega_p = \omega'_0 - \Omega_{eff} + \delta_{eff}$ is sent to the device around the red sideband of the HF cavity. A small VNA probe tone (black vertical arrow) with relative frequency $\Omega = \omega - \omega_p$ scans the HF reflection around $\omega'_0$. The beating of the pump and probe tones coherently drives the LF mode with a strength proportional to the dispersive coupling rate $g'_\omega$. The resulting LF amplitude in turn induces both a dispersively and a dissipatively generated sideband to the pump tone at $\omega_p + \Omega$, which are $\propto g'_\omega$ and $\propto ig'_\kappa/2$, respectively, and which both interfere with the original probe tone. **b** HF cavity reflection $|S_{11}|$ in the presence of a pump tone for three different pump detunings $\delta_{eff} = (-\kappa'_0/4, 0, +\kappa'_0/4)$ (from top to bottom). Data in left graphs are vs. detuning from HF cavity resonance $\omega - \omega'_0$, and clear PPIT signatures can be identified when $\omega - \omega'_0 \approx \delta_{eff}$. The smaller graphs

on the right show zooms to the transparency resonances (dashed boxes in left graphs) plotted vs. detuning from $\omega_p + \Omega_{eff}$. Symbols are data, lines are fits. The point of PPIT resonance in the zoom for $\delta_{eff} \approx 0$ is marked with an orange disk, and the cavity reflection for $g'_\omega = g'_\kappa = 0$ is added as dashed purple line in all zooms. **c** The dataset for $\delta_{eff} \approx 0$ in complex representation. Points are data, black line is a theory curve with all parameters from the fit and $\delta_{eff} = 0$. The cavity resonance traces a large circle anchored at (1, 0), the PPIT signature traces a smaller circle anchored on the resonance point of the bare cavity (purple disk); the resonance point of the PPIT is marked with the orange disk. For $g'_\kappa = 0$, the PPIT circle would point towards (1, 0) and the dashed line connecting the two resonance points would be the real axis. However, the presence of dissipative coupling leads to a rotation of the PPIT circle around its anchor point by an angle $\gamma = \arctan(g'_\kappa/2g'_\omega)$. Here, we obtain $\gamma \approx -46°$, which corresponds to $g'_\kappa/g'_\omega \approx -2.1$.

two I/O line pairs are connected to individual vector network analyzers (VNAs), and the HF input line is additionally connected to a microwave signal generator, which provides the sideband-pump tone. All experiments reported here were conducted with the sample directly immersed in liquid helium.

As first experiment, we characterize the reflection scattering matrix element $S_{11}$ of the two circuits via their individual feedlines around their respective resonance frequencies. From fits to the absorption dips, cf. Fig. 1d and "Methods—Resonance fitting", we extract the resonance frequencies of both modes (given above) as well as their internal and external decay rates $\kappa_{int} = 2\pi \times 38.2$ MHz, $\kappa_{ext} = 2\pi \times 6.7$ MHz for the HF circuit and $\Gamma_{int} = 2\pi \times 489$ kHz, $\Gamma_{ext} = 2\pi \times 36$ kHz for the LF circuit; data were taken at the flux sweetspot. Both circuits are undercoupled with $\kappa_{int}/\kappa_{ext} > 1$ and $\Gamma_{int}/\Gamma_{ext} > 1$. The high value for $\kappa_{int}$ can be attributed to the nano-constrictions, which have been shown to have a suppressed critical temperature and therefore considerable quasiparticle losses at 4.2 K[46,47]. However, this is not detrimental for the current experiment, but rather desired, since the high $\kappa_{int}$ is directly related to the dissipative coupling contribution. The large $\Gamma_{int}/\Gamma_{ext}$ has the advantage that the LF circuit does not need much attenuation in addition to the cable-attenuation to reach an effective mode temperature $T_{LF} \lesssim 10$ K, which is a big advantage if a cryogenic LF amplifier is not available (as was the case for our experiment).

## Flux-tunable photon-pressure interactions

The dispersive and dissipative single-photon coupling rates of a photon-pressure system are given by

$$g_{0\omega} = -\frac{\partial \omega_0}{\partial \Phi_b} \Phi_{zpf} \tag{1}$$

$$g_{0\kappa} = -\frac{\partial \kappa_0}{\partial \Phi_b} \Phi_{zpf}, \tag{2}$$

respectively, with the bias flux though the SQUID $\Phi_b$. Hence, to characterize the system and to select a good working point for the further experiments, as a next step we investigate the response of the HF resonance frequency and linewidth with respect to applied flux $\Phi_b$. We sweep the current through the attached coil, and for each current value we take a trace of both the HF and the LF reflection with the VNAs. Although the LF circuit is also weakly flux dependent (cf. Supplementary Note 3), we focus our discussion on the HF mode here, since its properties determine the interaction between the circuits. Our findings are summarized in Fig. 2.

Upon increasing $\Phi_b$ starting at the sweetspot $\Phi_b = 0$, we observe a redshift of the HF resonance frequency and an increase of the linewidth. Over a larger flux range, we observe that both quantities periodically modulate with period $\Phi_0$ as expected for a SQUID due to

fluxoid quantization. Note however, that $\omega_0$ and $\kappa_0 = \kappa_{int} + \kappa_{ext}$ modulate with opposite trend, i.e., when $\omega_0$ decreases $\kappa_0$ increases and vice versa. The maximum resonance frequency is found at the sweetspots $\Phi_b/\Phi_0 = n$ with $n \in \mathbb{Z}$, while this is the flux of the lowest decay rate. The modulation range for the resonance frequency is $\omega_0^{max} - \omega_0^{min} \approx 2\pi \times 43\,\text{MHz}$ and for the linewidth $\kappa_0^{max} - \kappa_0^{min} \approx 2\pi \times 81\,\text{MHz}$, i.e., $\kappa_0$ changes by about twice the amount $\omega_0$ does. From fits to the data points (for details see Supplementary Note 3), we determine the derivatives for both parameters to calculate the single-photon coupling rates $g_{0\omega}$ and $g_{0\kappa}$. Additionally, we obtain the ratio of the derivatives as a function of $\Phi_b$, which is equal to the flux-dependent ratio of the coupling rates $g_{0\kappa}/g_{0\omega}$.

As a good compromise between low circuit nonlinearity, medium total decay rate and maximum slope of both the tuning arcs, we choose the operation point $\Phi_b/\Phi_0 \approx 0.385$. Combining all parameters, we determine the dispersive and dissipative single-photon coupling rates at this flux point as $g_{0\omega} = 2\pi \times 27.1\,\text{kHz}$ and $g_{0\kappa} = 2\pi \times -48.8\,\text{kHz}$, respectively, which correspond to a large ratio $g_{0\kappa}/g_{0\omega} \approx -1.8$. Note that depending on the exact flux-bias point, this ratio can be tuned between $-1.2$ and $-3$, cf. Fig. 2d. The HF resonance frequency and linewidth at the working point are $\omega_0 \approx 2\pi \times 8.476\,\text{GHz}$ and $\kappa_0 \approx 2\pi \times 73.9\,\text{MHz}$; the LF parameters are $\Omega_0 \approx 2\pi \times 446.3\,\text{MHz}$ and $\Gamma_0 \approx 2\pi \times 601\,\text{kHz}$. The external decay rates are nearly constant as a function of flux, and so all the variation of $\kappa_0$ with $\Phi_b$ and $\Phi_{zpf}$ can be attributed to $\kappa_{int}$, cf. Supplementary Note 3. Note though, that in general it is important to discriminate between internal and external dissipative photon-pressure, since they can lead to qualitatively and quantitatively different consequences; here $g_{0\kappa_{int}} = g_{0\kappa}$ and $g_{0\kappa_{ext}} \approx 0$. Combined with the self-Kerr nonlinearity of the HF mode $\mathcal{K} \approx 2\pi \times -5.4\,\text{kHz}$ (the LF self-Kerr is negligibly small), cf. Supplementary Note 7, the device is well in the sideband-resolved regime with $\Omega_0/\kappa_0 \approx 6$ while in principle allowing for strong sideband pump tones due to $\mathcal{K}/\kappa_0 \ll 1$.

**Photon-pressure induced Fano-transparency**

To experimentally investigate the effects of the dissipative coupling contribution to the overall dynamics of the circuits, we begin with the protocol of photon-pressure induced transparency (PPIT). Here, a strong, fixed-frequency sideband-pump field is sent to the HF cavity around its red sideband $\omega_p = \omega_0' - \Omega_{eff} + \delta_{eff}$ with $|\delta_{eff}| \leq \kappa_0'/4$, and a much weaker VNA probe tone with frequency $\omega \sim \omega_0'$ is used to characterize the modified HF reflection response in a frequency span of few $\kappa_0'$ around $\omega_0'$, cf. Fig. 3a. We prime all HF mode quantities here to indicate that they shift with the intracavity pump photon number $n_c$ due to the Kerr nonlinearity and a nonlinear damping. The effective LF mode frequency $\Omega_{eff} = \Omega_0' + \delta\Omega_{pp}$ contains both, the power-shifted LF frequency due to a potential cross-Kerr effect $\Omega_0'$ and possible dynamical backaction contributions $\delta\Omega_{pp}$, which we do not know at this point and therefore choose $\Omega_{eff}$ as reference for the cavity red sideband. However, for all our data $\Omega_{eff} - \Omega_0 < \Gamma_0$, hence the difference is very small.

The beating of the pump and probe tones in the PPIT protocol coherently drives the LF circuit into oscillation, which in turn generates a sideband field to the pump tone, that interferes with the original probe tone with a phase-relation determined by the interaction and the detuning. As a consequence, a narrow transparency window with the shape of the effective LF susceptibility appears within the HF cavity resonance, a phenomenon closely related to EIT in atomic systems[48] and OMIT in optomechanical devices[49]. An interesting question is now whether and how the presence of dissipative coupling alters the usual, purely dispersive PPIT signature.

To model and understand the experiment, we derive the linearized equations of motion for the HF cavity field $\hat{c}$ and the LF mode field $\hat{d}$, which are given by

$$\dot{\hat{c}} = \left[-i\Delta_p' - \frac{\kappa_0'}{2}\right]\hat{c} - i\left[g_\omega' + i\frac{g_\kappa'}{2}\right]\left(\hat{d} + \hat{d}^\dagger\right) + i\sqrt{\kappa_{ext}'}\hat{c}_{in} \tag{3}$$

$$\dot{\hat{d}} = \left[i\Omega_0' - \frac{\Gamma_0'}{2}\right]\hat{d} - ig_\omega'\left(\hat{c} + \hat{c}^\dagger\right) \tag{4}$$

where $\Delta_p' = \omega_p - \omega_0'$, $\hat{c}_{in}$ represents the input probe tone, and $g_\omega'$ and $g_\kappa'$ are the dispersive and dissipative multiphoton coupling rates. We omit any input noise here, since it can be neglected to first order in a PPIT experiment, that is only considering the response to a coherent input tone. A complete derivation of the equations of motion can be found in Supplementary Notes 4 and 5.

One very interesting detail in the equations of motion is an asymmetry in the coupling terms. While the HF mode is coupled to the LF mode with a term $\propto g' = g_\omega' + ig_\kappa'/2$, only the dispersive interaction $g_\omega'$ contributes explicitly to changes of the LF mode field. In other words, the LF mode is driven proportional to $g_\omega'$ only, while the HF sideband is generated via $g_\omega' + ig_\kappa'/2$. By solving the equations of motion using Fourier transformation, combining the results, and applying the high-$Q$ limit for the LF mode (cf. Supplementary Note 5), we arrive at an expression for the HF reflection, when probed around resonance

$$S_{11} = 1 - \kappa_{ext}'\chi_c'\left[1 - g_\omega'\left(g_\omega' + i\frac{g_\kappa'}{2}\right)\chi_c'\chi_0^{eff}\right] \tag{5}$$

with the HF susceptibility $\chi_c' = [\kappa_0'/2 + i(\Delta_p' + \Omega)]^{-1}$, the effective LF susceptibility $\chi_0^{eff} = [\Gamma_0'/2 + i(\Omega - \Omega_0') + \Sigma]^{-1}$, the dynamical backaction $\Sigma = g_\omega'(g'\chi_{c0}' - g'^*\overline{\chi}_{c0}')$, $\chi_{c0}' = \chi_c'(\Omega_0')$, $\overline{\chi}_{c0}' = \chi_c'^*(-\Omega_0')$ and the probe frequency relative to the pump $\Omega = \omega - \omega_p$. Note that without dispersive coupling $g_\omega' = 0$ there would be no PPIT at all and no dynamical backaction, but both effects are nevertheless considerably modified by the presence of $g_\kappa'$ in $g'$. If we had an external-dissipative interaction instead of (or in addition to) the internal-dissipative one, the equations of motion and the reflection expression would contain additional terms and neither the dynamical backaction nor the PPIT signature would vanish for $g_\omega' = 0$[34].

As a consequence of $g_\kappa' \neq 0$, experiment and theory both reveal an interference-based and Fano-like modification of the PPIT resonance within the HF cavity resonance, cf. Fig. 3b. Most striking are two features. First, we have an asymmetric PPIT lineshape when the pump is directly on the effective red sideband $\delta_{eff} = \omega_p - (\omega_0' - \Omega_{eff}) \approx 0$ in contrast to dispersive coupling only. And secondly, the usual mirror symmetry for $+\delta_{eff}$ and $-\delta_{eff}$ is not preserved anymore. In the complex representation of $S_{11}$, cf. Fig. 3c, the origin of this additional tilt becomes apparent. Usually (i.e. for $g_\kappa' = 0$), the PPIT describes a small circle within the much larger HF cavity resonance circle, but both have their anchor points and their centers on the real axis for the pump on the red sideband. Due to the additional phase shift of $\pi/2$ of the dissipative coupling term ($i = e^{i\pi/2}$), however, the PPIT circle gets rotated around its anchor point.

Evaluating Eq. (5) at the corresponding frequency points (cf. "Methods−Fano-angle in transparency experiment" and Supplementary Note 6) reveals that in fact the angle $\gamma$ between the two circle axes is given by

$$\tan\gamma = \frac{g_\kappa'}{2g_\omega'}. \tag{6}$$

For the dataset discussed in Fig. 3, we find $\gamma = -46°$, which is equivalent to $g_\kappa'/g_\omega' \approx -2.1$. Observing this tilt is therefore not only a clear confirmation for the presence of internal-dissipative photon-pressure

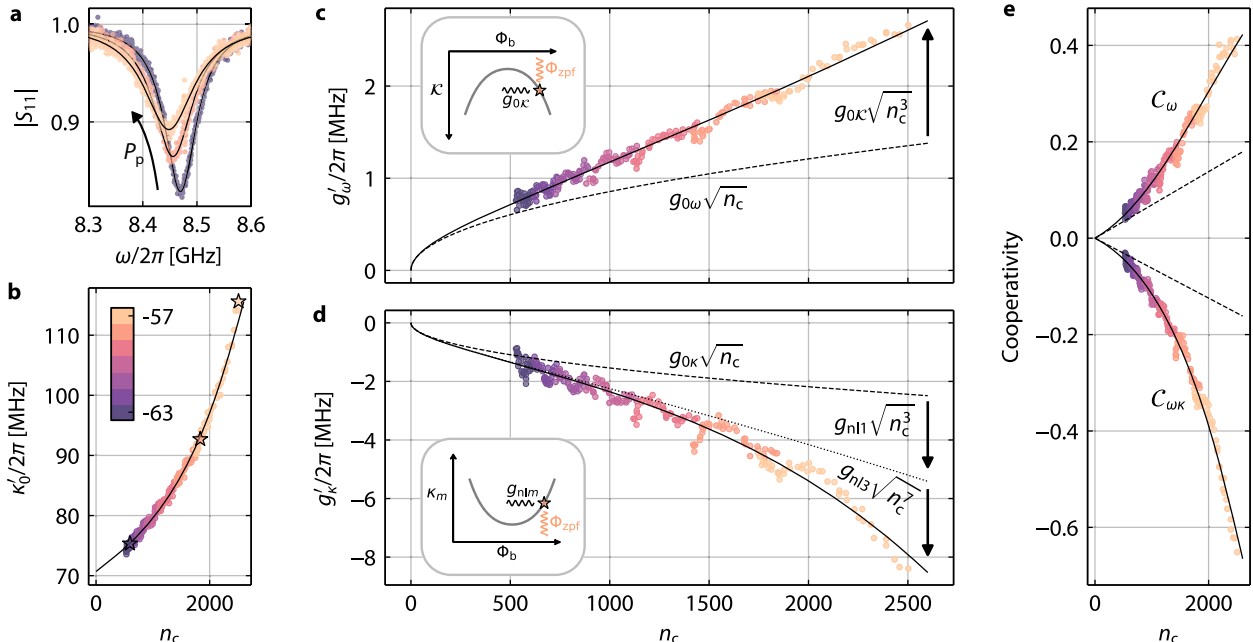

**Fig. 4 | Nonlinear damping and nonlinearity-enhanced photon-pressure coupling. a** HF cavity reflection $|S_{11}|$ for fixed $\omega_p = 2\pi \times 8.031$ GHz and increasing sideband pump-power $P_p$, measured by a small VNA probe tone. With increasing pump power, the cavity shifts to lower frequencies due to its Kerr anharmonicity, and its linewidth increases due to nonlinear damping. Symbols are experimental data, lines are fits. **b** Total HF mode linewidth $\kappa'_0$ for seven different pump powers (color-coded as given by the color bar in units of dBm) and multiple detunings between $-\kappa'_0/4$ and $+\kappa'_0/4$ in pump-frequency steps of 2 MHz, all plotted vs. their corresponding intracavity pump photon-number $n_c$. Symbols are data, star symbols correspond to the datasets in **a**. Line is a fit with $\kappa'_0 = \kappa_0 + 2\kappa_1 n_c + 3\kappa_2 n_c^2 + 4\kappa_3 n_c^3$. **c** Dispersive multiphoton coupling rate $g'_\omega$ vs. $n_c$ as extracted from PPIT. Pump powers and detunings are identical to **b**. Symbols are data, solid line is a fit with $g'_\omega = g_{0\omega}\sqrt{n_c} + g_{0\mathcal{K}}\sqrt{n_c^3}$; dashed line shows $g_\omega = g_{0\omega}\sqrt{n_c}$ without the Kerr enhancement. Inset: Schematic of the HF self-Kerr nonlinearity $\mathcal{K}$ as a function of

bias flux and how the LF zero-point-fluctuation flux modulates it by $g_{0\mathcal{K}} = -\Phi_{zpf}\partial\mathcal{K}/\partial\Phi_b$. **d** Dissipative multiphoton coupling rate $g'_\kappa$ vs. $n_c$ as extracted from PPIT. Pump powers and detunings are identical to **b**. Symbols are data, solid line is a fit with $g'_\kappa = g_{0\kappa}\sqrt{n_c} + g_{nl1}\sqrt{n_c^3} + g_{nl2}\sqrt{n_c^5} + g_{nl3}\sqrt{n_c^7}$; dashed line shows $g_\kappa = g_{0\kappa}\sqrt{n_c}$ without nonlinearity-enhancement, dotted line shows $g_{0\kappa}\sqrt{n_c} + g_{nl1}\sqrt{n_c^3}$. From the fit, we obtain $g_{nl2} \approx 0$. Inset: Schematic of a generic nonlinear damping coefficient $\kappa_m$, $m \in \mathbb{N}$, as a function of bias flux and how the LF zero-point-fluctuation flux modulates it by $g_{nlm} = -\Phi_{zpf}\partial\kappa_m/\partial\Phi_b$. **e** Dispersive cooperativity $\mathcal{C}_\omega = 4g'^2_\omega/\kappa'_0\Gamma_0$ and cross-cooperativity $\mathcal{C}_{\omega\kappa} = 2g'_\omega g'_\kappa/\kappa'_0\Gamma_0$ vs. $n_c$. Symbols are derived from data, solid lines follow from the fits in **b**–**d** with $\Gamma_0 = 2\pi \times 601$ kHz, dashed line is the reference without the nonlinearity-enhancement and with constant $\kappa'_0 = \kappa_0$. The effects from increasing $\kappa'_0$ and nonlinearity-enhanced $g'_\omega$, $g'_\kappa$ compete, but overall still lead to a significant enhancement of both cooperativities by factors up to ~2.4 and ~4.1, respectively.

coupling, but also might be a very useful and extraordinarily fast and simple method to quantify internal-dissipative coupling rates in the first place. It only requires a single complex resonance measurement and especially in devices, which do not provide the possibility to tune a system parameter $p$ to extract the derivatives of $\partial\omega_0/\partial p$ and $\partial\kappa_0/\partial p$ (e.g. in most optomechanical systems), the transparency rotation angle can be of high relevance. In this context it is also noteworthy, that external-dissipative photon-pressure does not result in the same Fano-interference in the sideband-resolved limit and with a pump tone around one of the sidebands, since its main consequence is a detuning-dependent re-scaling of $g'_\omega$ instead of adding an imaginary component to the total coupling rate[34].

The value of $g'_\kappa/g'_\omega \approx -2.1$ we find here from the PPIT data is only close to the single-photon equivalent $g_{0\kappa}/g_{0\omega} \approx -1.8$ obtained in the context of Fig. 2, but not exactly the same. The latter, i.e., $g_{0\kappa}/g_{0\omega} = g'_\kappa/g'_\omega$, would be expected if the multiphoton coupling rates scaled with pump photon number $n_c$ as usual as $g'_\omega = \sqrt{n_c}g_{0\omega}$ and $g'_\kappa = \sqrt{n_c}g_{0\kappa}$. The difference found here is not caused by inaccuracies, however, it rather points towards another highly interesting effect, which will be addressed in the next section.

### Nonlinearity-enhanced coupling rates

When performing the PPIT experiment with varying pump power and detuning, we observe that resonance frequencies, linewidths and coupling rates, and surprisingly even $g'_\kappa/g'_\omega$, depend on pump power.

The shift of $\omega'_0$ is explained by the Kerr anharmonicity of the HF mode, which stems from the nonlinear superconducting inductances. The origin of the nonlinear linewidth broadening, which is clearly faster than linear in pump photon number, cf. Fig. 4b, are ac-current-induced quasiparticles and an ac-current-induced suppression of the superconducting gap in the constrictions. We observe, consistent with that interpretation, that $\kappa'_{ext}$ is not significantly modified by the pump. The pump-broadened linewidth is phenomenologically modeled using $\kappa'_0 = \kappa_0 + 2\kappa_1 n_c + 3\kappa_2 n_c^2 + 4\kappa_3 n_c^3$ with $\kappa_0 = 2\pi \times 70.7$ MHz and the nonlinear-damping coefficients $\kappa_1 \approx 2\pi \times 3.7$ kHz, $\kappa_2 \approx 0$, and $\kappa_3 \approx 2\pi \times 0.37$ mHz; all values are obtained from a fit to the data. For the origin of the prefactors 2, 3 and 4 in the nonlinear contributions, cf. Supplementary Note 4.

Finally, we investigate $g'_\omega$ and $g'_\kappa$ as a function of pump photon number. A description of how we obtained the values from PPIT data is given in "Methods−Extracting the multiphoton coupling rates". In Fig. 4, we show all the multiphoton coupling rates obtained for multiple pump powers and detunings vs. $n_c$, and what we observe is surprising and exciting. Both coupling rates do not scale $\propto \sqrt{n_c}$ as expected from a simple first-order theory, but they increase considerably faster. While $g'_\omega$ seems to grow almost linearly, but with a finite value at $n_c = 0$, $g'_\kappa$ grows even faster, in particular at the highest photon numbers, similar to the increase of $\kappa'_0$ with $n_c$. For the lowest pump powers, the ratio $g'_\kappa/g'_\omega$ is very close to $g_{0\kappa}/g_{0\omega} = -1.8$, for the highest photon numbers though $g'_\kappa/g'_\omega \rightarrow -3.1$. The PPIT data in Fig. 3

**Table 1 | Nonlinearity coefficients for $\omega'_0$, $\kappa'_0$, $g'_\omega$ and $g'_\kappa$**

| $\mathcal{K}/2\pi$ | $\kappa_1/2\pi$ | $\kappa_2/2\pi$ | $\kappa_3/2\pi$ | $g_{0\mathcal{K}}/2\pi$ | $g_{\text{nl}1}/2\pi$ | $g_{\text{nl}2}/2\pi$ | $g_{\text{nl}3}/2\pi$ |
|---|---|---|---|---|---|---|---|
| − 5.4 | 3.7 | 0 | $3.7 \times 10^{-7}$ | 10.0 | − 22.1 | 0 | $- 3.5 \times 10^{-6}$ |

Values extracted from fits, numbers given in kHz for $\mathcal{K}$, $\kappa_1$, $\kappa_2$ and $\kappa_3$, and in Hz for $g_{0\mathcal{K}}$, $g_{\text{nl}1}$, $g_{\text{nl}2}$ and $g_{\text{nl}3}$.

are in the intermediate $g'_\kappa/g'_\omega$-regime (taken at $P_\text{p} = − 58$ dBm, i.e., $n_\text{c} \approx 1570$ for $\delta_\text{eff} \approx 0$), which explains the discrepancy between $g_{0\mathcal{K}}/g_{0\omega} = − 1.8$ and $g'_\kappa/g'_\omega = − 2.1$ found from $\gamma$.

Nonlinearity-enhanced multiphoton coupling rates are an intriguing and potentially very useful observation, which can be explained as follows. Each of the nonlinearity parameters $\mathcal{K}$, $\kappa_1$, $\kappa_2$ and $\kappa_3$ is flux-dependent[46], i.e., modulated by $\Phi_\text{zpf}$, and hence each can be the origin of an additional type of photon-pressure coupling. To first order, these additional coupling rates on the single-photon level are

$$g_{0\mathcal{K}} = - \frac{\partial \mathcal{K}}{\partial \Phi_\text{b}} \Phi_\text{zpf}, \quad g_{\text{nl}m} = - \frac{\partial \kappa_m}{\partial \Phi_\text{b}} \Phi_\text{zpf}, \quad m \in \mathbb{N} \quad (7)$$

and intrinsically they are several orders of magnitude smaller than $g_{0\omega}$ and $g_{0\kappa}$. In the framework of the linearized equations of motion, however, they get integrated into the multiphoton coupling rates as

$$g'_\omega = g_{0\omega}\sqrt{n_\text{c}} + g_{0\mathcal{K}}\sqrt{n_\text{c}^3} \quad (8)$$

$$g'_\kappa = g_{0\kappa}\sqrt{n_\text{c}} + g_{\text{nl}1}\sqrt{n_\text{c}^3} + g_{\text{nl}2}\sqrt{n_\text{c}^5} + g_{\text{nl}3}\sqrt{n_\text{c}^7}, \quad (9)$$

which means that despite their smallness on the single-photon level they can be very significant in the high-pump-power regime. Interestingly, also here we find a vanishing second order $g_{\text{nl}2} \approx 0$ from a corresponding fit, and for the finite coupling rates we get $g_{0\mathcal{K}} \approx 2\pi \times 10.0$ Hz, $g_{\text{nl}1} \approx 2\pi \times − 22.1$ Hz and $g_{\text{nl}3} \approx 2\pi \times − 3.5$ μHz. Table 1 summarizes all the relevant nonlinearity coefficients for $\omega'_0$, $\kappa'_0$, $g'_\omega$ and $g'_\kappa$.

For the Kerr constant, this effect has been predicted by theoretical work[50,51], and a related observation of much smaller magnitude has been reported in a multi-tone driven photon-pressure experiment[20]. The dissipative case, however, is even more impactful in our work than the dispersive one. While $g'_\omega$ is roughly doubled in the high-power regime compared to $g_{0\omega}\sqrt{n_\text{c}}$, which is already impressive, the dissipative multiphoton coupling $g'_\kappa$ gets boosted by up to a factor ~ 3.4 compared to $g_\kappa = g_{\kappa 0}\sqrt{n_\text{c}}$, cf. Fig. 4. Often, it is insightful to consider the cooperativity instead of the coupling rates, especially, when also the mode linewidth is a function of $n_\text{c}$ as is the case here. Unfortunately, the purely dissipative cooperativity $\mathcal{C}_\kappa = g'^2_\kappa/\kappa'_0\Gamma_0$ plays no role in this experiment due to the nonreciprocity of the interaction, but it would be enhanced by almost one order of magnitude. The dispersive cooperativity $\mathcal{C}_\omega = 4g'^2_\omega/\kappa'_0\Gamma_0$ and the cross-cooperativity $\mathcal{C}_{\omega\kappa} = 2g'_\omega g'_\kappa/\kappa'_0\Gamma_0$, on the other hand, are important for e.g. dynamical backaction (see below) and perspectively for sideband-cooling or parametric amplification. Those two cooperativities are still enhanced by a factor ~ 2.4 and ~ 4.1, respectively, which shows that the enhancement of the coupling rates by far overcompensates the increase of $\kappa'_0$.

Potentially, such cooperativity enhancements could lead to low-power groundstate-cooling or increased flux/displacement sensitivity, with a particular usefulness in optomechanics, where single-photon coupling rates are often very small compared to photon-pressure circuits and where device or setup heating by large pump powers is sometimes a limiting factor. Especially in superconducting circuits, it is furthermore possible to design $\mathcal{K}$ and correspondingly $\partial\mathcal{K}/\partial\Phi_\text{b}$ by carefully considered junction arrays[52], which might facilitate $g_{0\mathcal{K}}$ to be increased further by orders of magnitude in future systems.

The data in Fig. 4c and d also confirm once more, that the external-dissipative coupling rate plays no role here. If a significant contribution from $g_{\kappa_\text{ext}}$ was present in the device, this would lead to a dependence of the effective $g'_\omega$ on the detuning between pump and HF resonance $\Delta'^{34}_\text{p}$, which would have two notable consequences. First, not all the data for $g'_\omega$ in c would fall onto a single line, but within each color the values for high photon numbers would be pushed down (up), if $g_{0\kappa_\text{ext}} < 0$ ($g_{0\kappa_\text{ext}} > 0$), and the values for low photon numbers would be pushed up (down). And secondly, the ratio $g'_\kappa/g'_\omega$ would deviate considerably from the flux arc-based values even for very small $n_\text{c}$, cf. also Supplementary Note 6. Even though some scattering of both coupling rates is visible, none of the two effects is observed within the experimental accuracy.

## Dynamical backaction

Finally, we analyze the dynamical backaction exerted from the HF intracavity fields to the LF circuit as a function of power and detuning. This is not only an essential quantity for future experiments and applications like sideband-cooling with dissipative interactions, but an agreement with theoretical predictions will also serve as independent confirmation of our above conclusions, including the existence and strength of the dissipative coupling, the nonlinear enhancement of $\mathcal{C}_\omega$ and $\mathcal{C}_{\omega\kappa}$, and the photon-number dependent ratio of $g'_\omega/g'_\kappa$. Dynamical backaction arises from the time-delayed adjustments of the pump fields in the HF mode to changes of $\omega'_0$[17,53], here in our system caused by a change of flux through the SQUID. Hence, it depends on detuning between pump and cavity and on the cavity decay rate, which determines the time scale, on which this adjustment will happen. The effect can be separated into an in-phase and an out-of-phase component relative to the change of flux in the SQUID, which are equivalent to a change of effective LF frequency and a change of the LF decay rate.

Since $g'_\kappa$ comes with an additional phase-lag compared to $g'_\omega$, the dynamical backaction will likely be modified considerably in the presence of dissipative photon-pressure. And indeed, on a formal level we can directly see that the photon-pressure damping $\Gamma_\text{pp}$ and the photon-pressure frequency shift $\delta\Omega_\text{pp}$, which are given by

$$\Gamma_\text{pp} = 2\text{Re}\left[g'_\omega\left(g'\chi'_{c0} - g'^*\overline{\chi}_{c0}\right)\right] \quad (10)$$

$$\delta\Omega_\text{pp} = - \text{Im}\left[g'_\omega\left(g'\chi'_{c0} - g'^*\overline{\chi}_{c0}\right)\right] \quad (11)$$

deviate from the purely dispersive case, where the PP damping and PP frequency shift just encode the real and imaginary part of the cavity susceptibility, i.e., they are described by a complex Lorentzian. Strictly speaking, this is only completely true in the sideband-resolved regime, where $\chi'_{c0} - \overline{\chi}_{c0} \approx \chi'_{c0}$, but our device falls well within that approximation. With the phase-shifted dissipative coupling and the complex valued $g'$, the two backaction effects mix, i.e., they are rotated and scaled in the complex plane, which is no different to saying they acquire a Fano-like modification, very similar to what occurred in PPIT.

In Figure 5 we present the results of the experimental characterization of $\Gamma_\text{eff} = \Gamma'_0 + \Gamma_\text{pp}$ and $\Omega_\text{eff} = \Omega'_0 + \delta\Omega_\text{pp}$ and indeed find a significant deviation from the Lorentzian-like shape in the dispersive-only case. The data were acquired differently to the photon-pressure experiments discussed above. Here, we directly probe the LF mode via its own feedline, while stepping the HF red-sideband pump tone through various powers and detunings. The LF on-chip probe power was − 97 dBm, which − except for few data points very close to instability − is below the power required to observe LF nonlinearities, more details can be found in Supplementary Notes 7 and 8. One of the main advantages of this approach is the ability to investigate far more detunings, since, in contrast to the PPIT resonance, the LF mode visibility does not depend on the pump detuning and can be probed for

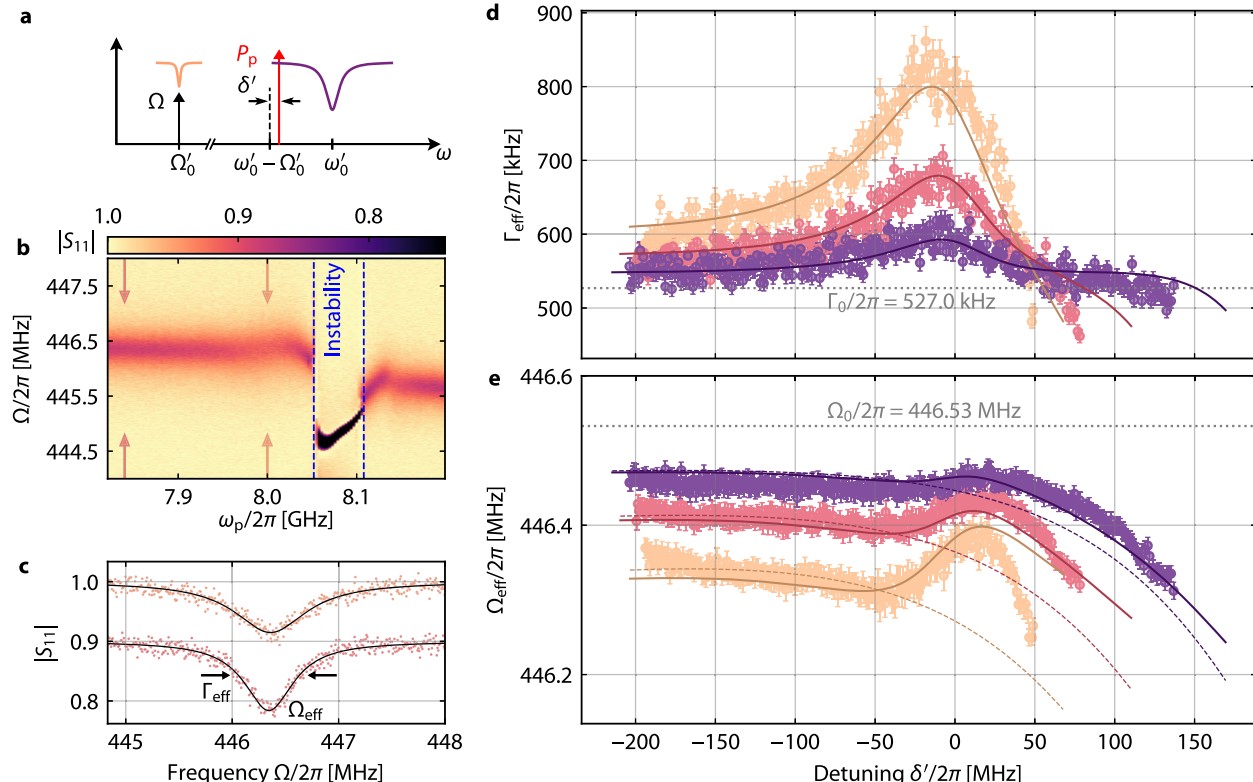

**Fig. 5 | Nonlinearity-enhanced dynamical backaction with dissipative coupling.**
**a** Schematic of the experiment. A pump tone (red arrow) with frequency $\omega_p$ and power $P_p$ is applied around the red sideband of the HF mode. Detuning of pump from the red sideband is $\delta' = \omega_p - (\omega_0' - \Omega_0')$. For each pair $(\omega_p, P_p)$, the LF mode reflection is probed directly around $\Omega_0'$ using the LF feedline and a VNA (black arrow). **b** Color-map of the LF reflection $|S_{11}|$ vs. pump frequency $\omega_p$. When the pump is far red-detuned from the HF mode red-sideband, the resonance is nearly unmodified compared to the pump-free case. Just below $\delta' = 0$ or $\omega_p/2\pi \approx 8.0$ GHz, the absorption gets wider and shallower and slightly shifts in frequency; both effects indicate dynamical backaction at work. Strikingly, for $\omega_p/2\pi \approx 8.05$ GHz, which is still around the cavity red-sideband, a parametric instability region appears, which is experimentally identified by a strongly deformed resonance feature in $S_{11}$, and the simultaneous disappearance of the HF mode in the HF reflection[54]. The two pairs of arrows indicate the linescans shown in **c**. Top curve in **c** shows $|S_{11}|$ at $\omega_p/2\pi \approx 8.0$ GHz, the point of maximum photon-pressure damping,

bottom curve is manually offset by $-0.1$ for clarity and is for $\omega_p/2\pi \approx 7.84$ GHz. Symbols are data, lines are fits, from which we extract $\Omega_{eff}$ and $\Gamma_{eff}$. **d, e** Effective LF mode linewidth $\Gamma_{eff}$ and effective resonance frequency $\Omega_{eff}$ vs. detuning $\delta'$ for three different $P_p$, color code for $P_p$ identical to Fig. 4, cf. Supplementary Fig. 15 for the corresponding $n_c$. Symbols are data and error bars are standard errors obtained from the fit routine, lines are theory curves. Note the two clearest signatures for dissipative coupling: first, the maximum for the photon-pressure damping is at negative detunings from the red sideband. And second, for positive detunings from the red sideband the total damping rate is smaller than the intrinsic damping rate $\Gamma_{eff} < \Gamma_0$, where $\Gamma_0$ is shown as dashed line. This indicates negative backaction damping with a red-detuned pump, and is the precursor for the approaching parametric instability. To fully model the data, we included a cross-Kerr frequency shift to the LF frequency $\Omega_0' = \Omega_0 + \mathcal{K}_c n_c$ with $\mathcal{K}_c = 2\pi \times -130.8$ Hz, and a small nonlinear cross-damping $\Gamma_0' = \Gamma_0 + \kappa_c n_c$ with $\kappa_c = 2\pi \times 38.4$ Hz. Dashed lines in **e** show $\Omega_0'$.

small pump photons numbers $n_c$, i.e., for large detunings between HF cavity and pump.

A non-intuitive result of these experiments is that we observe a parametric instability, when the pump has a slightly higher frequency than $\omega_0' - \Omega_0'$, but is still far red-detuned. This is also predicted by theory, cf. Supplementary Note 6, and it is an intrinsic signature for the presence of dissipative coupling, which has also been studied in related optomechanical systems[24,33]. We can understand it from Eq. (10), which includes (for simplicity again in the sideband-resolved limit) that the photon-pressure damping $\Gamma_{pp} = g_\omega'^2 \kappa_0' |\chi_{c0}|^2 + g_\omega' g_\kappa' \delta' |\chi_{c0}|^2$ has two contributions, the second of which changes sign with the pump detuning from the red sideband $\delta' = \omega_p - (\omega_0' - \Omega_0')$ and is $< 0$ for $\delta' > 0$. So it enhances the dispersive PP damping for $\delta' < 0$, and counteracts it for $\delta' > 0$. In our case it even overcompensates it to $\Gamma_{eff} \leq 0$ (theoretical instability criterion) around $\delta' \approx +\kappa_0'$ due to the large $|g_\kappa'/g_\omega'|$ ratio for large $n_c$. Similar considerations are valid for $\delta\Omega_{pp}$. An additional signature for backaction interference is that the maximum of $\Gamma_{eff}$ occurs at $\delta' < 0$, cf. Fig. 5d, instead of at $\delta' \approx 0$ as expected for $g_\kappa' = 0$. The slight shift of the maximum $\Gamma_{eff}$ to lower frequencies with increasing $P_p$ on the other hand reflects the increasing ratio $|g_\kappa'/g_\omega'|$. A

more detailed comparison between purely dispersive and mixed photon-pressure can be found in Supplementary Note 6.

To quantitatively model $\Gamma_{eff}$ and $\Omega_{eff}$, we include also a cross-nonlinear damping and a cross-Kerr frequency shift, i.e.,

$$\Gamma_{eff} = \Gamma_0 + \kappa_c n_c + \Gamma_{pp} \tag{12}$$

$$\Omega_{eff} = \Omega_0 + \mathcal{K}_c n_c + \delta\Omega_{pp}, \tag{13}$$

where both nonlinear effects are occurring naturally due to the constrictions being part of the LF mode in the galvanic coupling scheme. We fit the backaction data using all the HF mode and photon-pressure parameters obtained independently, and with the free parameters $\Gamma_0$, $\Omega_0$, $\mathcal{K}_c$ and $\kappa_c$. The cross-parameters we obtain from this are rather small with $\mathcal{K}_c = 2\pi \times -130.8$ Hz and $\kappa_c = 2\pi \times 38.4$ Hz, but the shift due to $\mathcal{K}_c$ is still dominating the total pump-induced LF frequency shift. The cross-nonlinear damping on the other hand only contributes less than 10% to the deviations of $\Gamma_{eff}$ from $\Gamma_0$ at the maximum of $\Gamma_{eff}$, and the biggest part is due to $\Gamma_{pp}$. With the cross-nonlinearities included,

we find good agreement between theory and data. The remaining deviations we attribute to uncertainties in the frequency-dependent pump-photon number, higher-order nonlinearities for the largest photon numbers and the onset of the parametric instability for the points close to the instability threshold. The recovery of the LF mode from the instability regime at $\omega_p/2\pi \gtrsim 8.1$ GHz, which is visible in Fig. 5b but not predicted by theory, is accompanied by the absence of the HF mode in the HF reflection[54], which we believe is due to the pump photon number being so large that the cJJ SQUID is driven into the voltage state by overcritical peak HF currents. Overall though, the agreement between backaction data and model clearly confirms our above conclusions regarding Fano-like effects by dissipative coupling contributions and the nonlinearity-enhanced coupling rates, without which the datasets would lie much closer together for the pump powers used in Fig. 5d, e.

## Discussion

In this work, we have reported the realization of niobium-based superconducting photon-pressure circuits, which are operated at liquid helium temperature in the highly dissipative circuit regime. Due to the elevated temperature and dissipation compared to earlier implementations, we obtained a device with a considerable flux-tunable dissipative interaction $g_{0\kappa}$ in addition to the usual dispersive coupling $g_{0\omega}$, and with $-1.2 \gtrsim g_{0\kappa}/g_{0\omega} \gtrsim -3$. Furthermore, we observed at least three additional photon-pressure coupling terms due to the flux-modulation of inherent device nonlinearities, namely a Kerr-anharmonicity interaction and both a first- and a third-order nonlinear-damping-based interaction. The additional coupling terms were shown to lead to a strong multiphoton enhancement of the original interaction terms by up to a factor 3.4 and of the cooperativities by up to a factor 4.1. We have revealed with both our experiments and our theoretical description that the presence of dissipative photon-pressure leads to a nonreciprocal interaction between the modes, and to a Fano-like distortion of photon-pressure induced transparency due to the phase-shifted interference of the two coupling types, which can be used to determine $g_\kappa/g_\omega$ from a single response trace in frequency. A related Fano-like distortion is present in the frequency-dependence of dynamical backaction, when $g_\kappa \neq 0$, which leads to enhanced photon-pressure damping when the sideband-pump is red-detuned from the cavity red-sideband and to a counter-intuitive instability when the pump tone is placed between the red cavity sideband and the cavity resonance.

In summary, our results reveal and describe several unexplored types of interaction between two superconducting circuits and demonstrate a series of experimental consequences of the presence of a purely-internal-dissipative radiation-pressure coupling. This work opens the door for the investigation and application of dissipative photon-pressure in circuit QED, for unexplored low-frequency photon control protocols, and for photon-pressure experiments in the thermal and dissipative regime due to its compatibility with liquid helium. Such tools are relevant for research fields like dark matter axion detection, since they provide new possibilities for LF photon-sensing and control, potentially down to the kHz frequency range. Our report also confirms that photon-pressure circuits are an ideal testbed for general radiation-pressure systems. It is directly relevant for and applicable to optomechanical systems, in particular to SQUID optomechanics, where similar effects can be expected to be observed and utilized in the future.

Finally, the experiments presented here suggest interesting questions and directions for future developments. The most immediate question is how the dissipative contribution modifies sideband-cooling. Generally, it will furthermore be very interesting to lower the LF mode resonance frequency to investigate the unresolved-sideband regime and corresponding effects like cooling on resonance or novel squeezing protocols, since in this regime dissipative coupling is predicted to have its biggest strengths. Engineering an external-dissipative coupling by a modulation of $\kappa_{ext}$ remains an open challenge, that would provide an additional degree of freedom and would bring new dynamics into play, since in such a system the LF mode directly couples to the pump tone on the HF feedline, while simultaneously not necessarily requiring a low $Q$ of the HF mode. Once such an external-dissipative coupling can be realized, i.e., a flux-dependent coupling between the HF mode and a CPW feedline, the same approach could be used to implement effective internal-dissipative couplings even in the case of flux-independent constriction losses, e.g. at mK temperatures, by using a second "internal" feedline or on-chip resistor which is coupled to the SQUID circuit in a flux-dependent fashion. Lastly, the observed nonlinearity-enhancements suggest that it might be worth to target stronger or even different types of non-linearities both in photon-pressure circuits and in optomechanics.

## Methods
### Device fabrication
*Step 1: Bottom niobium layer.* The fabrication starts with dc-magnetron-sputtering of 120 nm thick niobium (Nb) on top of a high-resistivity ($\rho > 10$ k$\Omega$ cm at room temperature) intrinsic two inch silicon wafer. The wafer has a thickness of 525 µm, and after Nb deposition it is covered with a positive photoresist (ma-P 1205) by means of spin coating (resist thickness ~ 0.5 µm). The bottom niobium layer design is transferred to the photoresist using maskless photolithography ($\lambda_{litho} = 365$ nm). The irradiated resist is developed with ma-D 331/S for 25 s, and then the excess Nb film is removed by $SF_6$ reactive ion etching. Finally, the wafer is cleaned with multiple subsequent baths of acetone and isopropanol.

*Step 2: Dielectric layer.* The second step is to deposit a dielectric layer on the patterned first layer of Nb, where the parallel-plate-capacitors (PPCs) will be formed. Before deposition, the wafer is covered with the same photoresist as in step 1, and the areas where the PPCs are formed are defined again by maskless optical lithography. After the resist is developed, the wafer is completely covered by plasma-enhanced chemical vapor deposition (PECVD) with 200 nm of silicon nitride ($Si_3N_4$) in a low-temperature process ($T$ ~ 100°C). Then, a lift-off procedure is performed in acetone for 15 min to remove the resist and all the excess $Si_3N_4$. Finally, the wafer is rinsed in multiple baths of acetone and isopropanol.

*Step 3: Top niobium layer.* The third step in the fabrication procedure is similar to step 2, but instead of $Si_3N_4$, 300 nm of Nb is deposited onto the resist-covered and patterned wafer, again by dc-magnetron sputtering. After lift-off of the excess Nb in acetone, the second Nb layer forms the PPC top plate, the transmission line of the low-frequency (LF) resonator, and the ground planes of both high-frequency (HF) and LF resonators.

*Step 4: Mounting and pre-characterization.* After fabrication, the wafer is diced into individual $10 \times 10$ mm$^2$ large chips. Then, a single chip is mounted into a matching cutout of a microwave printed circuit board (PCB), and is wirebonded to the microwave feedlines and ground plane of the PCB. Both chip and PCB are packed in a radiation-tight copper housing. Finally, the chip is mounted into the measurement setup, and the device is pre-characterized in liquid helium by means of microwave reflectometry. Here, each of the modes is characterized via its own CPW feedline.

*Step 5: Constriction fabrication.* To cut the constriction Josephson junctions (cJJs) into the pre-characterized device, the sample is removed from the PCB and mounted onto an aluminum stub; it is wirebonded to the stub to prevent any charging of the sample during the ion irradiation. The mounted sample is placed in the neon ion microscope, which allows high-precision milling with a focused neon ion beam (Ne-FIB). The cJJs are of the monolithic 3D-type[46,47]. They are formed by cutting two ~ 40 nm narrow slot-shaped rectangles from both sides into the 3 µm wide bridges with a dose of 20000 ions/nm$^2$

and an accelerating voltage of 20 kV. In addition, the constrictions are milled from the top with a third rectangle, but with a lower dose of 3000 ions/nm².

*Step 6: Mounting and experiments.* After the Ne-FIB cutting process, the sample is mounted again into the measurement setup, similar to step 4, but this time a small coil for the application of a magnetic field perpendicular to the chip surface is added. Then, the photon-pressure experiments begin.

## Resonance fitting

In this Methods part, we describe the fitting routine, which we used to fit all pumped and unpumped HF and LF resonances as well as the PPIT zoom measurements. We will generically use the unprimed HF quantities here $\omega_0$, $\kappa_{\text{ext}}$, $\kappa_{\text{int}}$, and $\kappa_0$ to do so, but the same formalism is valid for the primed quantities and the capitalized LF quantities. Some more details and a corresponding figure can be found in Supplementary Note 7.

The ideal reflection-response function of a high-$Q$ ($Q \gg 10$) parallel RLC circuit, which is capacitively coupled to a feedline with characteristic impedance $Z_0$, is given by

$$S_{11}^{\text{ideal}} = 1 - \frac{2\kappa_{\text{ext}}}{\kappa_0 + 2i(\omega - \omega_0)} \tag{14}$$

with the angular excitation frequency $\omega$, the corresponding resonance frequency $\omega_0$, the external linewidth $\kappa_{\text{ext}}$ and the total linewidth $\kappa_0 = \kappa_{\text{int}} + \kappa_{\text{ext}}$.

Due to the cabling and all the microwave components in between the vector network analyzer and the circuit, the ideal transmission is not what we measure though. To take frequency-dependent attenuation, the electrical cable length and possible interferences (e.g. parasitic transmission through the directional coupler or parasitic reflections) into account, we model the actual reflection as

$$S_{11}^{\text{real}} = (a_0 + a_1\omega + a_2\omega^2)\left(1 - \frac{2\kappa_{\text{ext}}e^{i\theta}}{\kappa_0 + 2i(\omega - \omega_0)}\right)e^{i(\phi_0 + \phi_1\omega)}. \tag{15}$$

The factors $a_0$, $a_1$, $a_2$, $\phi_0$, $\phi_1$ and $\theta$ are real-valued fit parameters. Before we apply this equation to the large-linewidth HF data, however, we divide the experimental $S_{11}^{\text{exp}}$ data by a corresponding high-power background dataset $S_{11}^{\text{bg, exp}}$, that has been taken in a VNA power-regime, in which the resonance is completely suppressed due to its nonlinearity. As a result, we obtain $S_{11}^{\text{cor}} = S_{11}^{\text{exp}}/S_{11}^{\text{bg, exp}}$, details are described in Supplementary Note 7. Afterwards, the background reflection is sufficiently smooth to apply Eq. (15).

During our automated data fitting routine we first remove the absorption resonance from the dataset (leaving a gapped $S_{11}$-dataset) and fit the remaining $S_{11}$-response in an appropriate frequency window (typically five to ten times $\kappa_0$) with the background function

$$S_{11}^{\text{bg}} = (a_0 + a_1\omega + a_2\omega^2)e^{i(\phi_0 + \phi_1\omega)}. \tag{16}$$

We obtain preliminary values for $a_0$, $a_1$, $a_2$, $\phi_0$ and $\phi_1$. Then, we calculate $S_{11}^{\text{cor}}/S_{11}^{\text{bg}}$ for the complete dataset and fit the resulting data with

$$S_{11}^{\theta} = 1 - \frac{2\kappa_{\text{ext}}e^{i\theta}}{\kappa_0 + 2i(\omega - \omega_0)}, \tag{17}$$

from which we obtain a preliminary set of values for $\omega_0$, $\kappa_0$, $\kappa_{\text{ext}}$ and $\theta$. Finally, we use all the preliminary values for $a_0$, $a_1$, $a_2$, $\phi_1$, $\phi_2$, $\omega_0$, $\kappa$, $\kappa_{\text{ext}}$ and $\theta$ as starting parameters to re-fit the original dataset with the complete Eq. (15). Here, we find that $\theta$ is small and nearly constant and during the last step, we keep $\theta = \theta_0 = 0.08$ constant to improve the quality of the fits and the values of the remaining parameters. All the

HF $S_{11}$-datasets after constriction cutting, which are shown in the manuscript and Supplementary Material figures, as well as their corresponding fit curves have been background-corrected using this method. Additionally, we have rotated off the interference angle $\theta_0$.

## Fano-angle in transparency experiment

The reflection response of the PPIT experiment is modeled using Eq. (5) (for the derivation see Supplementary Note 5)

$$S_{11} = 1 - \kappa_{\text{ext}}'\chi_c'\left[1 - g_\omega'g'\chi_c'\chi_0^{\text{eff}}\right] \tag{18}$$

with the multi-photon coupling rates $g_\omega'$, $g_\kappa'$ and $g' = g_\omega' + i\frac{g_\kappa'}{2}$, the HF cavity susceptibility

$$\chi_c' = \frac{1}{\frac{\kappa_0'}{2} + i(\Delta_p' + \Omega)} \tag{19}$$

and the effective LF susceptibility

$$\chi_0^{\text{eff}} = \frac{1}{\frac{\Gamma_{\text{eff}}}{2} + i(\Omega - \Omega_{\text{eff}})}. \tag{20}$$

Exactly on resonance of both HF cavity and PPIT, i.e., when $\Delta_p' + \Omega = \Omega - \Omega_{\text{eff}} = 0$ (imaginary parts of both susceptibilities vanish) or alternatively $\Delta_p' = -\Omega_{\text{eff}}$, we get for the reflection at $\Omega = +\Omega_{\text{eff}}$

$$S_{11}^{\text{res}} = 1 - 2\frac{\kappa_{\text{ext}}'}{\kappa_0'}\left[1 - 4\frac{g_\omega'g'}{\kappa_0'\Gamma_{\text{eff}}}\right]. \tag{21}$$

For vanishing coupling $g' = g_\omega' = 0$, the resulting cavity resonance point is the usual one

$$S_{11}^{\text{HF}} = 1 - 2\frac{\kappa_{\text{ext}}'}{\kappa_0'}, \tag{22}$$

i.e., exactly on the real axis.

To find the angle between the real axis and the connecting line between that point and the PPIT resonance, we shift the PPIT reflection by this value and get

$$S_{11}^{\text{res, shift}} = 8\frac{\kappa_{\text{ext}}'}{\kappa_0'}\frac{g_\omega'g'}{\kappa_0'\Gamma_{\text{eff}}}. \tag{23}$$

The angle is then obtained via

$$\gamma = \arctan\frac{\text{Im}[S_{11}^{\text{res, shift}}]}{\text{Re}[S_{11}^{\text{res, shift}}]} \tag{24}$$

$$= \arctan\frac{g_\kappa'}{2g_\omega'}. \tag{25}$$

## Extracting the multiphoton coupling rates

To study the scaling of the multiphoton coupling rates with intracavity pump photon number, we perform the experiment described in Fig. 3 for several different pump powers and detunings in the range $\delta_{\text{eff}} \in [-\kappa_0'/4, \kappa_0'/4]$. The PPIT data and fits for the resonant cases $\delta_{\text{eff}} \approx 0$ are presented for all $P_p$ in Supplementary Note 8. The range of detunings within the $\kappa_0'/2$-wide interval around the red sideband is chosen to ensure a clear PPIT signature for all $\delta_{\text{eff}}$ and $P_p$, while simultaneously avoiding parametric instabilities. For the analysis, we first fit the cavity resonance only of each dataset by removing the narrow PPIT window from it and obtain as fit parameters $\omega_0'$ and $\kappa_0'$, which fully determine $\chi_c'$; additionally, we obtain $\kappa_{\text{ext}}'$. Secondly, we fit

the isolated high-resolution PPIT resonance in its narrow frequency span and obtain as fit parameters $\Gamma_{\text{eff}}$ and $\Omega_{\text{eff}}$, i.e., the effective LF linewidth and resonance frequency including dynamical backaction effects, which together determine $\chi_0^{\text{eff}}$. In a last step, we fit the combined cavity-plus-PPIT data using Eq. (5) with all parameters fixed from the individual fits except for $g'_\omega$ and $g'_\kappa$. The deviations of $g'_\omega$ and $g'_\kappa$ from the fit lines in Fig. 4 seem to increase with $|\delta_{\text{eff}}|$, and we attribute them to a frequency-dependence of the cavity background parameters such as $\theta$, which are not completely accounted for.

The intracavity pump photon number $n_c$, which is required to analyze $g'_\omega(n_c)$ and $g'_\kappa(n_c)$, we obtain from the experimental ac Stark shift of the HF mode $\delta\omega_0 = \omega'_0 - \omega_0$ in combination with the theoretical value for $g_{0\omega}$. A detailed description is presented in Supplementary Notes 4 and 7.

### Fit approach for dynamical backaction

To model the dynamical backaction curves in Fig. 5, we use Eqs. (10) and (11) to calculate $\Gamma_{\text{pp}}$ and $\delta\Omega_{\text{pp}}$ without any free parameters. The scaling of $g'_\omega, g'_\kappa$ and $\kappa'_0$ with $n_c$ we obtain from the corresponding fits shown in Fig. 4, while the HF mode frequency $\omega'_0$ as a function of $n_c$ we calculate as

$$\omega'_0 = \omega_p + \sqrt{(\Delta_p - \mathcal{K}n_c)(\Delta_p - 3\mathcal{K}n_c) - \frac{\kappa_{\text{aux}}^2}{4}} \tag{26}$$

and $\kappa_{\text{aux}} = \kappa_1 n_c + 2\kappa_2 n_c^2 + 3\kappa_3 n_c^3$, cf. Supplementary Notes 4 and 7. The only missing ingredient then is $n_c(\omega_p)$, which we calculate by numerically solving the characteristic polynomial of a superconducting circuit with a dispersive Kerr nonlinearity and up to third order nonlinear damping

$$\frac{\kappa_3^2}{4}n_c^7 + \frac{\kappa_2\kappa_3}{2}n_c^6 + \frac{\kappa_2^2 + 2\kappa_1\kappa_3}{4}n_c^5$$
$$+ \frac{\kappa_0\kappa_3 + \kappa_1\kappa_2}{2}n_c^4 + [\mathcal{K}^2 + \frac{\kappa_1^2 + 2\kappa_0\kappa_2}{4}]n_c^3 \tag{27}$$
$$+ [\frac{\kappa_0\kappa_1}{2} - 2\mathcal{K}\Delta_p]n_c^2 + [\Delta_p^2 + \frac{\kappa_0^2}{4}]n_c - \kappa_{\text{ext}}n_{\text{in}} = 0$$

cf. Supplementary Note 4. The input photon flux

$$n_{\text{in}} = \mathcal{G}_{\text{att}}(\omega_p)\frac{P_{\text{sg}}}{\hbar\omega_p} \tag{28}$$

contains the experimentally determined pump-attenuation $\mathcal{G}_{\text{att}}(\omega_p)$ and the output power of the signal generator $P_{\text{sg}}$, cf. Supplementary Note 7.

Finally, we fit the total effective linewidth and resonance frequency for all pump powers simultaneously, using Eqs. (12) and (13) with $\Omega_0$, $\Gamma_0$, $\kappa_c$ and $\mathcal{K}_c$ as fit parameters. Notably, the fit values for $\Gamma_0$ and $\Omega_0$ obtained from this fit deviate a bit from the ones obtained from the flux arc fits given in "Results—Flux-tunable photon-pressure interactions", cf. also Supplementary Note 4. To be more precise, they differ by ~70 kHz and ~190 kHz, respectively. A good part of the deviation in $\Omega_0$ can be attributed to the LF flux arc fit (and the data extracted from it) slightly deviating from the data points, which accounts for ~100 kHz in $\Omega_0$. We believe the remaining mismatches are mainly due to a 5 dB difference in probe powers used for the LF reflection measurements during the flux arc sweep (higher probe power) and the dynamical backaction experiment (lower probe power). The latter would also be consistent with probe-power dependence measurements of the LF circuit during another cooldown, where we observed slight shifts of $\Omega_0$ and $\Gamma_0$ as function of probe power even below the onset of significant self-Kerr nonlinearities.

## Data availability

All data presented in this paper and the Supplementary Material (including raw data) and the corresponding processing scripts used during the analysis are publicly available on the repository Zenodo[54].

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

## Acknowledgements

The authors thank Markus Turad, Ronny Löffler (instrument scientists of the core facility LISA+), and Christoph Back for technical support. Furthermore, we thank Monika Fleischer and Ralf Stiefel for providing access to the PECVD system. This research received funding from the Deutsche Forschungsgemeinschaft (DFG) via grant numbers 490939971 (BO 6068/1-1, recipient D.B.) and 511315638 (BO 6068/2-1, recipient D.B.). M.K. gratefully acknowledges financial support by the Studienstiftung des deutschen Volkes, J.P. acknowledges support from the Cusanuswerk, Bischöfliche Studienförderung. We also gratefully acknowledge support by the COST actions NANOCOHYBRI (CA16218) and SUPERQUMAP (CA21144). The authors finally acknowledge support by the Open Access Publishing Fund of the University of Tübingen.

## Author contributions

M.K. designed and fabricated the device, carried out the experiments, performed data analysis, prepared the figures and contributed to the first draft of the manuscript. J.P. performed data analysis, prepared the figures, and contributed to the circuit design and to the first draft of the manuscript. Z.E.G. contributed to fabrication recipe, data acquisition, data analysis, figure preparation, and to the first draft of the manuscript. B.W. developed the measurement code and contributed to theory. K.U. contributed to the device fabrication. D.K. and R.K. contributed to the project funding and participated in scientific discussions. D.B. conceived the experiment, supervised all parts of the project, acquired the project funding, developed the theoretical framework and wrote the first

draft of the manuscript. All authors discussed the results and conclusions and contributed to manuscript revisions.

## Funding

## Competing interests

The authors declare no competing interests.
