## [Transparent Peer Review file · Nature Communications]

Tunable and nonlinearity-enhanced dispersive-plus-dissipative coupling in photon-pressure circuits

Corresponding Author: Mr Mohamad Adnan El Kazouini

Version 0:

Reviewer comments:

Reviewer #1

(Remarks to the Author)

In this manuscript, the authors report the realization of niobium-based superconducting photon-pressure circuits operated at liquid helium temperatures in a highly dissipative regime. The work discusses not only dissipative coupling but also Kerr-anharmonic interactions, as well as first- and third-order nonlinear damping processes.

The manuscript is well written and presents a detailed and thoughtful discussion of the measurements and results. The demonstrated effect is significant and could open promising avenues for exploring the unresolved-sideband regime, with implications for cooling on resonance and novel squeezing protocols—regimes where dissipative coupling is expected to show its greatest advantages. I support publication in Nature Communications after the authors address my comments.

From the plots in Supplemental Fig. 5, it appears that the contribution of the dissipative coupling in the extrinsic channel is about $g_{ke} = 2\pi \times 4.9\text{kHz}$. Although this value is roughly ten times smaller than the intrinsic dissipative coupling rate, it enters the rate/coupling equations differently, since it also modulates the input field for both the HF and LF modes. Have the authors considered this contribution explicitly? How would it affect the interpretation of the results? Relatedly, since the g_{ke} term might play a role by modulating the input power into the system, this could influence several of the reported observations. In particular, when ω_p is near $\delta=0$, the nonlinearity of the system could produce a driving force near the low-frequency mechanical mode. This may suggest that the observed instability point -- and the discrepancy between the data and suggested model -- corresponds to a synchronization mechanism, such as injection locking of the down-converted HF pump frequency to the LF mode, mediated by the same nonlinear term. Similar effects are known to depend strongly on pump power through the nonlinear coupling [1,2,3,4].

[1] Appl. Phys. Lett. 93, 191115 (2008)

[2] Opt. Express 26, 8275–8288 (2018)

[3] Phys. Rev. Applied 12, 024034 (2019)

[4] Nat. Commun. 12, 5625 (2021)

Minor comments

(i) In Supplementary Fig. 8, why is Γ_{pp} not symmetric for $g_k=0$?

(ii) Why was the experiment not performed systematically near the largest values of g_k and $g\omega$? From Fig. 2 this point should be close to $\phi_b/\phi_0=0.5$

(iii) At times, the notation becomes difficult to follow due to the large number of constants and rates. For example, in the nonlinear section I could not find a clear table defining all constants such as $\kappa_{int,2}$, $\kappa_{ext,2}$, $\kappa_{int,4}$, $\kappa_{ext,4}$, etc. It would be very helpful if the authors included a comprehensive table summarizing these constants and explicitly showing how they add up to the effective rates k_1 , k_2 ,... etc., used in the main text.

(iv) I am curious to see a photograph of the experimental setup, particularly the copper enclosure and liquid helium implementation, to help assess how scalable this solution may be.

Reviewer #2

(Remarks to the Author)

In the manuscript titled “Tunable and nonlinearity enhanced dispersive-plus-dissipative coupling in photon pressure circuits” the authors realize a Niobium based superconducting device where a high frequency photonic mode is coupled to a low frequency photonic mode via photon pressure interaction which is analogous to the radiation pressure coupling in cavity

optomechanical systems. The coupling stems from the modulation of flux threading the SQUID loop due to the zero-point current of the LF mode.

In addition to the dispersive coupling g_{ω} between the modes the authors report observation of dissipative coupling g_{κ} where primarily the internal dissipation rate of the HF mode is modulated. The dissipative coupling modifies the photon pressure induced transparency (PPIT) into a fano-like response. By fitting the PPIT data, the coupling rates are estimated to be as high as 3.4. Further the authors observe the nonlinear enhancement of the multi-photon coupling rates, which deviates from the usual $\sqrt{n_c}$ scaling with photon number.

Additionally, the authors show that the strong dissipative coupling leads to a modification in the dynamical backaction, resulting in an enhanced photon-pressure damping and parametric instability of the LF mode under near-red-sideband pump in the HF mode.

The manuscript is well written and all the key results are highlighted clearly. The fabrications details, theoretical models, data analysis and fitting routines are described thoroughly. The work is an important step towards understanding the potential of nonlinearity enhanced multi photon coupling-particularly dissipative coupling and opens pathways for future exploration in nonlinear quantum optomechanics in photon-pressure circuits and related systems. Therefore, I recommend this work to be published in Nature communication once the authors address the following points.

1. What modification occurs in the LF mode backaction under blue-sideband drive? Moreover, how does the backaction evolve (at least theoretically) in the entire range from red to blue sideband drive?
2. How was the LF mode instability condition defined?
3. What input power was used to probe the LF mode to measure the dynamical backaction data in Fig. 5? What is the corresponding average number of photons in the LF mode? Was it sufficiently low enough to not affect the backaction?
4. In the fig. 5b it is observed that beyond the pump frequency 8.1 GHz the LF mode's instability vanishes, whereas in the theoretical curves (fig. 5 d and supplementary fig 8.a) the instability persists with higher positive detuning δ^{\wedge} . How far the instability extends theoretical and can the authors explain the mismatch?
The authors might consider to look into the Ref.10.1103/PhysRevLett.114.043601 (cited in the manuscript), where the return to "normalcy" from instability show good agreement with the theory, to find out the potential missing element in their own theoretical modelling.
5. Was there any specific reason to use directional coupler instead of a circulator for the reflection measurement?
6. High temperature bath and relevance- The authors mention in the discussion that the elevated temperature acted in favor of high dissipation rate for HF mode leading to larger dissipative coupling. However, one typically aims to lower the system temperature to mitigate quasiparticle loss and initialize HF mode in quantum ground state to observe interesting quantum effects. How a large dissipative coupling be realistically achieved in mK temperatures regime? If not, how would that impact the relevance or applicability of the present study to future quantum experiments?
7. What was the on-chip pump power and corresponding photon number at $\delta^{\wedge}=0$ during dynamical backaction measurement?
8. As per my understanding the authors demonstrate a fano feature in PPIT/OMIT response with a red sideband for the first time which is particularly interesting. The authors provide a detailed theoretical analysis in supplementary note V (F, I) and, VI (B) which explains the origin of fano response. Previous optomechanical systems (e.g. ref.34) observe OMIT arising from dissipative coupling but without such fano features. What enables the authors to observe such fano PPIT response? Is it primarily due to stronger g_{κ} or some additional contributing factor?
9. Regarding SQUID optomechanics, I suggest that the authors additionally cite the following relevant works for completeness and context: 10.1038/s41598-022-05438-x and 10.1038/s42005-020-00514-y
10. Distinction of internal vs external decay modulation: The dissipative coupling reported here arises from modulation of internal decay rate. Can the authors highlight the results that specifically depends on κ_i modulation which would otherwise be not present with κ_{ext} modulation? Additionally, can similar effects be achieved in circuits that exhibit external dissipative coupling?

Version 1:

Reviewer comments:

Reviewer #1

(Remarks to the Author)

The authors have comprehensively addressed the concerns raised in my initial critique, as well as those posed by the other reviewer. Given the improvements evident in the resubmitted manuscript and the enhanced details provided within the

supplementary information, I am now satisfied and recommend the paper for publication in Nature Communications.

Reviewer #2

(Remarks to the Author)

In the revised manuscript the authors have successfully addressed all the queries of reviewers. I am satisfied with additional sections regarding the external dissipative coupling and possible explanation of the mismatch between data and theory of Fig. 5.

Therefore, I recommend publication of work in Nature communication journal.

Point-by-Point Reply to the Reviewer Reports

Note:

All section, figure and paragraph numbers as well as references in the reply are given with respect to *the revised version* of the manuscript and the Supplementary Material. Line numbers specifically are given with respect to the *diff* file.

Reply to the comments of reviewer #1

Reviewer #1:

“In this manuscript, the authors report the realization of niobium-based superconducting photon-pressure circuits operated at liquid helium temperatures in a highly dissipative regime. The work discusses not only dissipative coupling but also Kerr-anharmonic interactions, as well as first- and third-order nonlinear damping processes.

The manuscript is well written and presents a detailed and thoughtful discussion of the measurements and results. The demonstrated effect is significant and could open promising avenues for exploring the unresolved-sideband regime, with implications for cooling on resonance and novel squeezing protocols – regimes where dissipative coupling is expected to show its greatest advantages. I support publication in Nature Communications after the authors address my comments.”

Reply:

We thank the reviewer for their careful reading of our manuscript, the concise summary of its content, their appreciation of its quality, and for their recommendation to publish the work in Nature Communications after addressing their helpful comments. We will happily do so.

Reviewer #1:

“From the plots in Supplemental Fig. 5, it appears that the contribution of the dissipative coupling in the extrinsic channel is about $g_{ke} = 2\pi \cdot 4.9\text{kHz}$. Although this value is roughly ten times smaller than the intrinsic dissipative coupling rate, it enters the rate/coupling equations differently, since it also modulates the input field for both the HF and LF modes. Have the authors considered this contribution explicitly? How would it affect the interpretation of the results?”

Reply:

We agree with the reviewer, an extrinsic-dissipative coupling would enter the equations differently and lead for example to direct coupling between the LF mode and the pump tone on the HF feedline and to modified HF output fields. Since in principle this can lead to very interesting additional effects, we are currently working on a new circuit design, which will enable us to controllably activate a tunable extrinsic-dissipative interaction.

We are not certain how the reviewer determined the 4.9 kHz from Supplementary Fig. 6. Based on the fit curve of κ_{ext} shown there, the extrinsic-dissipative single-photon coupling rate at the operation point would only be $g_{0\kappa_{\text{ext}}}/2\pi \approx -0.55\text{ kHz}$, i.e., two orders of magnitude smaller than its intrinsic counterpart, so $g_{0\kappa_{\text{ext}}}/g_{0\kappa_{\text{int}}} \approx 10^{-2}$. This can also be eyeballed from the two orders of magnitude difference in y -axis scaling between Supplementary Fig. 6b and d, but visually nearly identical fit-curve slopes at the operation point. And although this is very small, it would still not be completely negligible but lead to e.g. a modified effective g'_{ω} with an explicit dependence on the detuning between sideband pump and HF cavity, which should be observable in the experiment if $g_{\kappa_{\text{ext}}}$ was significant, but is not, cf. main paper Fig. 4.

We are convinced the reason for the absence of indications for external-dissipative coupling is that the change of κ_{ext} , which is visible in Supplementary Fig. 6, is an artifact of the fit routine and the considerably undercoupled cavity, and that the actual extrinsic-dissipative coupling rate is more than one order of magnitude smaller than what the fit suggests. In this case, it can safely be neglected. For the detailed explanations and corresponding discussions, we kindly refer the reviewer to the extensive new paragraphs and sections in the revised Supplementary Material as listed next.

Change to manuscript:

We have added new discussions on the apparent flux-tuning of κ_{ext} and estimates on $g_0\kappa_{\text{ext}}$ in Supplementary Note III D (lines 168-183) and Supplementary Note III E (lines 212-222). Furthermore and also in the context of a comment by reviewer #2, we have added a completely new Supplementary Note VI D (lines 545-597), which explicitly and formally considers the impact of an extrinsic-dissipative coupling to the effective coupling rates, dynamical backaction and PPIT.

Reviewer #1:

“Relatedly, since the g_{ke} term might play a role by modulating the input power into the system, this could influence several of the reported observations. In particular, when ω_p is near $\delta=0$, the nonlinearity of the system could produce a driving force near the low-frequency mechanical mode. This may suggest that the observed instability point – and the discrepancy between the data and suggested model – corresponds to a synchronization mechanism, such as injection locking of the down-converted HF pump frequency to the LF mode, mediated by the same nonlinear term. Similar effects are known to depend strongly on pump power through the nonlinear coupling [1,2,3,4].

[1] Appl. Phys. Lett. 93, 191115 (2008)

[2] Opt. Express 26, 8275–8288 (2018)

[3] Phys. Rev. Applied 12, 024034 (2019)

[4] Nat. Commun. 12, 5625 (2021)”

Reply:

We thank the reviewer for this interesting input. However, we are not certain how injection locking could lead to the observed instability. From our understanding, injection locking requires an additionally injected signal, to which the frequency and phase of a parametric oscillator – often one that performs already self-sustained oscillations – then lock. Also it seems that in the four references provided by the reviewer the oscillator is first driven to self-sustained oscillations by a parametric instability, and then a second input signal is injected into the device, to which the pre-existing self-sustained oscillations are locking. Without this additional tone, to which reference should the LF oscillator synchronize? Similarly to the description in the four references, in our system the instability observed in Fig. 5 occurs independently of the LF probe signal or any other additionally injected signal. We can also observe it by using a spectrum analyzer to passively listen to the HF mode output field with only the pump tone applied. And in particular for $\delta' = 0$ we observe significant backaction damping, which roughly speaking is the opposite of an intrinsic feedback driving force. Nevertheless, the reviewer’s comment got us thinking – what might actually be possible from our point of view is that within the instability region the self-sustained oscillations of the LF mode lock to the VNA probe signal. This seems indeed interesting to look into in a future experiment.

Beyond that, we believe that the current interpretation of the observed instability is satisfying. It is predicted by the theory of dispersive-plus-dissipative coupling, and the remaining discrepancies between theory and experiment are qualitatively consistent with the intuitive expectations for higher-order nonlinearities, overcritical peak microwave currents in the nano-constrictions for very high pump powers, and some unavoidable inaccuracies in the parameters entering the equations (e.g. pump photon numbers). We would also like to refer again to our new Supplementary Note VI D (lines 545-597),

which – in agreement with other work (e.g. Ref. [34]) – demonstrates that the potential impact of a small external-dissipative coupling would be to re-scale the effective coupling rates and introduce a detuning-dependence in g'_{ω} , but would not lead to fundamentally different phenomena.

It is worth noting here that we were able to somewhat improve the agreement between theory and data for the dynamical backaction by refitting the LF reflection data with a nonlinear circuit model, since we discovered in the context of a comment by reviewer #2, that at the points close to the instability small signatures of an LF self-Kerr nonlinearity are present in the LF S_{11} traces, which were neglected so far.

Change to manuscript:

We have reduced the discrepancy between theory and experimental data for the dynamical backaction (in particular for Ω_{eff} close to instability) by taking the self-Kerr nonlinearity of the HF mode (relevant close to instability) into account during data analysis, cf. Supplementary Notes IV A to IV C (lines 224-250) as well as the new Note VII C (lines 676-687), and finally the updated data in main paper Fig. 5 and Supplementary Fig. 17.

Reviewer #1:

“Minor comments

(i) In Supplementary Fig. 8, why is Γ_{pp} not symmetric for $g_k=0$?”

Reply:

The backaction curve is not symmetric in Supplementary Fig. 9, since for the shown curves we keep the pump power constant as a function of the detuning, not the intracavity pump photon number n_c . As a consequence, n_c increases with decreasing pump detuning from the cavity (i.e. from left to right in the plot) and so does the backaction, which is $\propto n_c$.

Although we stated the detuning-independence of the pump power and the resulting detuning-dependence of n_c already in both the caption and the corresponding text in the Supplementary Material of the original manuscript, we decided to add an even more explicit statement on the asymmetry of the $g_k = 0$ curve to the caption of Supplementary Fig. 9 and to the corresponding text.

Note, however, that even for $n_c = \text{const.}$ the backaction curves would only look symmetric on the scale chosen there, since we do not apply the sideband-resolved limit for the corresponding curves, and when the pump moves closer to resonance with the HF mode, the backaction will deviate from a Lorentzian tail and cross 0 when the pump is exactly on resonance. The latter can be seen in the new Supplementary Fig. 11, which was triggered by a comment of reviewer #2 and which shows the backaction for all detunings from $-1.5\Omega_0$ to $+1.5\Omega_0$ for constant n_c .

Change to manuscript:

We added an explicit statement on the asymmetry of the $g_k = 0$ curve to the captions of Supplementary Fig. 9 and the new Supplementary Fig. 10 as well as to the text in line 437. We also repeat the discussion in the context of the new Supplementary Fig. 11 (lines 489-491).

Reviewer #1:

“(ii) Why was the experiment not performed systematically near the largest values of g_k and g_{ω} ? From Fig. 2 this point should be close to $\Phi_b/\Phi_0=0.5$ ”

Reply:

The reason for the chosen operation point – as stated in the main manuscript in lines 224 to 226 – is that it provides a good compromise between low HF mode Kerr nonlinearity, medium HF mode linewidth and large coupling strengths. Moving closer to $\Phi_b/\Phi_0 = 0.5$ is possible, but it significantly increases the cavity linewidth, cf. Fig. 2c, which makes the resonance dip broader and shallower (κ_{ext} is nearly

constant, while κ_{int} increases to ≥ 110 MHz) and therefore complicates data analysis and cavity fitting, especially in the high-pump-power regime, where the internal HF linewidth increases even further due to the nonlinear damping.

In future devices we might choose a larger external linewidth κ_{ext} , which will facilitate moving to a larger bias flux while still having a clear and deep resonance dip.

Reviewer #1:

“(iii) At times, the notation becomes difficult to follow due to the large number of constants and rates. For example, in the nonlinear section I could not find a clear table defining all constants such as $\kappa_{\text{int},2}$, $\kappa_{\text{ext},2}$, $\kappa_{\text{int},4}$, $\kappa_{\text{ext},4}$, etc. It would be very helpful if the authors included a comprehensive table summarizing these constants and explicitly showing how they add up to the effective rates k_1 , k_2 ,... etc., used in the main text.”

Reply:

We agree with the reviewer and are grateful for the idea of a parameter table. In fact, we liked it so much that we added two, a small one summarizing the values for the nonlinearity parameters in the main manuscript and a long one defining the most important variables and their most relevant relations to other variables at the very beginning of the Supplementary Material. However, the external coupling rate is in good approximation a constant of pump power, so $\kappa_{\text{ext},m} = 0$ and $\kappa_m = \kappa_{\text{int},m}$ for all $m > 0$, while $\kappa_0 = \kappa_{\text{int}} + \kappa_{\text{ext}}$. Similarly $g_{\kappa_{\text{ext}}} = 0$ and $g_{\text{nlm},\text{ext}} = 0$.

Change to manuscript:

We have included a new table summarizing the relevant nonlinearity parameters into Sec. II D of the main manuscript (page 7, bottom right), and a more extensive table collecting and describing the most important variables of the manuscript on page 3 of the Supplementary Material.

Reviewer #1:

“(iv) I am curious to see a photograph of the experimental setup, particularly the copper enclosure and liquid helium implementation, to help assess how scalable this solution may be.”

Reply:

We are not certain what the reviewer envisions here in terms of scalability, but we are happy to include some photographs of the setup and the chip mount.

Change to manuscript:

We have included a new Supplementary Fig. 2, showing the copper sample mount and the relevant end part of the liquid helium dipstick as it was mounted during the experiments. A corresponding new sentence in the main text of the Supplementary Material can be found in lines 7-8.

Reply to the comments of reviewer #2

Reviewer #2:

“In the manuscript titled “Tunable and nonlinearity enhanced dispersive-plus-dissipative coupling in photon pressure circuits” the authors realize a Niobium based superconducting device where a high frequency photonic mode is coupled to a low frequency photonic mode via photon pressure interaction which is analogous to the radiation pressure coupling in cavity optomechanical systems. The coupling stems from the modulation of flux threading the SQUID loop due to the zero-point current of the LF mode.

In addition to the dispersive coupling g_{ω} between the modes the authors report observation of dissipative coupling g_{κ} where primarily the internal dissipation rate of the HF mode is modulated. The dissipative coupling modifies the photon pressure induced transparency (PPIT) into a fano-like response. By fitting the PPIT data, the coupling rates are estimated to be as high as 3.4. Further the authors observe the nonlinear enhancement of the multi-photon coupling rates, which deviates from the usual $\sqrt{n_c}$ scaling with photon number.

Additionally, the authors show that the strong dissipative coupling leads to a modification in the dynamical backaction, resulting in an enhanced photon-pressure damping and parametric instability of the LF mode under near-red-sideband pump in the HF mode.

The manuscript is well written and all the key results are highlighted clearly. The fabrications details, theoretical models, data analysis and fitting routines are described thoroughly. The work is an important step towards understanding the potential of nonlinearity enhanced multi photon coupling-particularly dissipative coupling and opens pathways for future exploration in nonlinear quantum optomechanics in photon-pressure circuits and related systems. Therefore, I recommend this work to be published in Nature communication once the authors address the following points.”

Reply:

We thank the reviewer for their careful reading of our manuscript, the concise summary of its content, their appreciation of its importance and quality, and for their recommendation to publish the work in Nature Communications after addressing their valuable comments. We will happily do so.

Reviewer #2:

“1. What modification occurs in the LF mode backaction under blue-sideband drive? Moreover, how does the backaction evolve (at least theoretically) in the entire range from red to blue sideband drive?”

Reply:

Due to time constraints we unfortunately did not systematically characterize the dynamical backaction with a pump tone around the HF blue sideband or around HF resonance. In between the manuscript measurements and today, the device also considerably changed its properties (we suspect the cJJs are aging) and so we cannot ‘complete’ the datasets anymore.

However, we are more than happy to add the corresponding plots of the theoretical data and their discussion to the Supplementary Material.

Change to manuscript:

We added the new Supplementary Figs. 10 and 11 as well as the corresponding discussions (lines 472-500) to Supplementary Note VI A.

Reviewer #2:

“2. How was the LF mode instability condition defined?”

Reply:

The theoretical condition is defined as usual by $\Gamma_{pp} \leq -\Gamma'_0$, i.e., by the total LF damping rate being zero or negative $\Gamma_{\text{eff}} = \Gamma'_0 + \Gamma_{pp} \leq 0$, cf. Supplementary Material line 442 and the captions of Supplementary Figs. 9, 10 and 11. The experimental LF instability condition was not defined in a similarly strict sense. What we observe in the region labeled “instability” in Fig. 5b is accompanied by multiple phenomena indicating large-amplitude self-sustained LF oscillations:

- At the threshold linescans marking the transitions from the instability region to the usual region and back to “normalcy”, we observe (likely probe-tone or flux-noise triggered) abrupt jumps between two states of the LF circuit in S_{11} .
- Within the instability region itself, the reflection does not look at all like a usual Lorentzian resonance anymore, but rather shows a considerably deformed, and much narrower and deeper resonance feature (also indicated by the black color in Fig. 5, which is not artificially added, but shows that the depth of the feature exceeds the color scale by far).
- At the start of the instability region at $\omega_p/2\pi \sim 8.05$ GHz, the HF resonance abruptly disappears from the HF reflection, likely pushed to low frequencies and large linewidths by the large-amplitude self-sustained LF oscillations.
- Finally, in a subsequent experiment using a spectrum analyzer and measuring the HF output field, we observed a narrow and strong signal in the power spectral density in the instability region as typical for upconverted self-sustained oscillations (since for those datasets device characteristics and operation point were different from the current manuscript, we do not include them for consistency).

Potentially, we might also have injection locking in this regime, when probing the LF mode with a VNA (at least when the probe-frequency is close to Ω'_0), as suggested by reviewer #1, which may lead to frequency dragging in the black-colored region of Fig. 5b and the specific shapes of the deformed resonances there. However, due to all the nonlinearities in the system, it is not straightforward to model the self-sustained oscillations and to understand the details of the data and the modified responses. Nevertheless, we are considering to look into this regime again in the future in a dedicated experiment due to the large interest by both reviewers.

Change to manuscript:

We have added a clarifying statement about the experimental signatures of the instability to the caption of main paper Fig. 5 and a short statement on the theoretical criterion in main paper lines 526–527.

Reviewer #2:

“3. What input power was used to probe the LF mode to measure the dynamical backaction data in Fig. 5? What is the corresponding average number of photons in the LF mode? Was it sufficiently low enough to not affect the backaction?”

Reply:

The first two questions can be easily answered and incorporated into the revised manuscript. Our estimate is that the LF on-chip probe power for the data presented in Fig. 5 was $P_\beta \approx -97$ dBm. The intracircuit LF probe photon number then depends on detuning from the LF resonance and the dynamical backaction damping, but for the backaction-free parameters $\Omega_0/2\pi \approx 446$ MHz, $\Gamma_0/2\pi \approx 600$ kHz and $\Gamma_{\text{ext}}/2\pi \approx 37$ kHz the maximum photon number on resonance is $\lesssim 45000$.

The answer to the third question seems more intricate. We believe the reviewer refers to a kind of pump depletion here, in which case we expect this would be indicated by the LF resonances deviating from Lorentzian lineshapes, since the reduction of the dynamical backaction would be a function of the probe photon number, which in turn depends on LF probe detuning from the LF resonance. Upon a careful re-inspection of the LF reflection datasets, we could indeed find that close to the parametric instability (the last ~ 10 data points for each pump power) the LF resonance shows a small but visible Duffing tilt, which slipped our attention before and hence was not considered in the evaluation. We are not sure whether this is just a usual LF self-Kerr effect or indeed a signature of pump depletion, but we adjusted the fitting routine to include the self-Kerr resonance deformation and to provide us with the unshifted Ω_{eff} as a fit parameter. This approach indeed improved the agreement between the experimental and the theoretical Ω_{eff} close to instability (and left all other values essentially unchanged), cf. main paper Fig. 5 and Supplementary Fig. 17, and we are particularly grateful to the reviewer for bringing our attention in this direction.

For all datasets, which show symmetrical and linear LF resonances, we are convinced that the probe power is low enough not to impact the dynamical backaction.

Change to manuscript:

We have included a statement on the LF probe power to the main text lines 504-507, which also refers to Supplementary Notes VII and VIII for further details. In the Supplementary Material, we have expanded the equation of motion for the LF mode by a self-Kerr nonlinearity in Notes IV A to IV C, and we have adjusted the fitting routine for the backaction datasets to take this self-Kerr nonlinearity into account. Naturally, this led to an updated main paper Fig. 5 and an updated Supplementary Fig. 17. A description of the adjusted fitting routine is given in the new Supplementary Note VII C (lines 676-687). In Supplementary Note VIII B (lines 762-767), we again state the on-chip probe power, this time including the corresponding intracircuit LF probe photon number.

Reviewer #2:

“4. In the fig. 5b it is observed that beyond the pump frequency 8.1 GHz the LF mode’s instability vanishes, whereas in the theoretical curves (fig. 5 d and supplementary fig 8.a) the instability persists with higher positive detuning δ^* . How far the instability extends theoretical and can the authors explain the mismatch?”

The authors might consider to look into the Ref.10.1103/PhysRevLett.114.043601 (cited in the manuscript), where the return to “normalcy” from instability show good agreement with the theory, to find out the potential missing element in their own theoretical modelling.”

Reply:

Indeed, these are sharp observations and very good questions by the reviewer. In a simple theory without HF mode nonlinearities, which is what is plotted in Supplementary Fig. 9 and the new Supplementary Figs. 10 and 11, the instability would extend all the way to the blue sideband for constant pump power. Similarly, if we include the HF nonlinearities which are already considered in the manuscript (first order Kerr, third order nonlinear damping). However, for much larger pump photon numbers than our maximum of ~ 2500 , i.e., for the pump tone being closer to resonance or larger P_p , we expect higher-order nonlinearities and overcritical microwave peak currents through the nano-constrictions to become dominant (a slight onset of such a higher order dissipative contribution can already be suspected from the last data points of Figs. 4b and d); we are convinced that these are the missing elements explaining the specific “return to normalcy” (RTN) observed in Fig. 5.

What we observe in particular in the RTN region $\omega_p/2\pi \geq 8.15$ GHz is that the HF mode has disappeared from the HF reflection, while the LF reflection looks almost like in the case of no pump, just a bit red-shifted. Our interpretation is that the microwave currents due to the pump tone are so large that

the SQUID is driven to a voltage-state during the peak currents of the HF oscillation. Such switching would “kill” the HF mode, but not the LF mode, which still has a low impedance path around the constrictions, cf. main paper Fig. 1. This conclusion can be supported by a rough quantitative estimate: the peak microwave currents in the HF mode are given by $I_p \approx 2I_{zpf}^{HF} \sqrt{n_c}$, and for an estimated pump photon number of $n_c \approx 7000 - 8000$ at $\omega_p/2\pi \approx 8.15$ GHz (extrapolated from the data in Supplementary Fig. 15 for the highest power), we get $I_p \approx 170I_{zpf}^{HF} \approx 15 \mu\text{A}$ with $I_{zpf}^{HF} \approx 88$ nA. The critical current of the SQUID on the other hand we estimate to be $I_0 \approx 25 \mu\text{A}$ at the operation point, which is still a bit higher but on the same order of magnitude. From earlier work on Ne-FIB cJJs (e.g. Refs. [48,49]), however, we also know that the switching current I_{sw} to the voltage state of this type of nano-constrictions is typically $I_{sw} \approx 0.75I_0$, which here would correspond to $I_{sw} \approx 19 \mu\text{A}$. Hence, I_p and I_{sw} are sufficiently similar to be consistent with our interpretation, in particular when considering that in reality one of the cJJs might have a lower critical current than the second, and other contributions like flux noise might lower the effective I_{sw} further. A final aspect supporting this conclusion is that the width of the instability region actually increases for decreasing pump power.

The experiment and theory in the paper suggested by the reviewer Ref. [33] is in a quite different regime than our work. The system only has an external-dissipative coupling, no intrinsic Kerr nonlinearity or nonlinear damping, no cavity breakdown for high pump-photon numbers and it is well in the bad cavity limit. Interestingly, however, and likely for a different reason, even in this manuscript, there are no data for the RTN region in the lower-right panel of Fig. 4, only a theory line.

Change to manuscript:

We added a brief statement regarding the RTN for $\omega_p/2\pi \gtrsim 8.1$ GHz to the main text (lines 556-562), which includes our interpretation of it as switching of the cJJs to the voltage state by peak microwave currents.

Reviewer #2:

“5. Was there any specific reason to use directional coupler instead of a circulator for the reflection measurement?”

Reply:

We agree with the reviewer, in principle we could have also used a circulator for the reflection measurement. The directional coupler has several advantages though. First and most importantly, the directional coupler is somewhat smaller (at least in the critical direction) and hence it is easier to fit two of them simultaneously into our limited experimental space. Also, directional couplers do not contain magnetic materials, which might cause offset magnetic fields and magnetic noise in the vicinity of the SQUID. As a bonus: directional couplers are typically more affordable.

Reviewer #2:

“6. High temperature bath and relevance- The authors mention in the discussion that the elevated temperature acted in favor of high dissipation rate for HF mode leading to larger dissipative coupling. However, one typically aims to lower the system temperature to mitigate quasiparticle loss and initialize HF mode in quantum ground state to observe interesting quantum effects. How a large dissipative coupling be realistically achieved in mK temperatures regime? If not, how would that impact the relevance or applicability of the present study to future quantum experiments?”

Reply:

This is a relevant question raised by the reviewer. At this point, we cannot say how exactly the intrinsic-dissipative coupling rate of this specific device will change when it is operated at mK, since we have not tested it. But we expect it to be much lower, since both κ_{int} and the change of κ_{int} with bias flux

should be considerably reduced, cf. also Refs. [48,49] for the characteristics of similar cJJ circuits at temperatures below 4.2 K (not mK though). On the other hand it is also true that the observation of interesting quantum effects will most likely require the HF mode to be in the quantum ground state. There is a way out, though. As mentioned in the context of other questions raised by both reviewers, we are currently working on a concept and design to realize external-dissipative coupling using a flux-dependent coupling element between the HF mode and its coplanar waveguide feedline. With the same approach, one could then design large and flux-tunable internal coupling rates as well, if instead of to the actual CPW feedline the HF mode is coupled to a simple resistor by means of a flux-dependent coupler. Hopefully, the reviewer will understand that at this point we prefer not to disclose the layout and conceptual details behind this approach. Instead, to show that in principle it is possible, we would like to refer to the paper Y. Yin *et al.*, Phys. Rev. Lett. **110**, 107001 (2013), where a strongly flux-tunable and SQUID-based coupling between a microwave resonator and a waveguide feedline has already been implemented. Replacing the feedline with an on-chip resistor or terminating the feedline with an off-chip one (and at the same time adding an actual drive and readout line at another spot) will then make the internal decay rate a highly designable parameter independent of temperature. Other circuit modifications using dedicated resistive elements within the circuits are also possible.

In the current experiment, we get the large internal-dissipative coupling “for free” due to the high-temperature, which considerably simplifies its basic experimental investigation. All the interesting (yet classical) consequences we observe, such as Fano transparency or modified dynamical backaction, on the other hand are not dependent on the actual temperature and hence their observation and understanding is relevant for all related systems with internal-dissipative radiation-pressure, also for future ones at mK temperature or for optomechanical systems in the optical domain (where the cavity is always in the ground state).

Change to manuscript:

We have added a corresponding statement into the main paper discussion section (lines 629-636), and briefly mentioned the topic again in the context of designing the sign of g_k in the Supplementary Material (lines 498-500).

Reviewer #2:

“7. What was the on-chip pump power and corresponding photon number at $\delta' = 0$ during dynamical backaction measurement?”

Reply:

Indeed, this information was not explicitly contained in the original manuscript. The only place showing all $n_c(\omega_p)$ is Supplementary Fig. 15c, however, as a function of ω_p and not of δ' , and so the information for $\delta' = 0$ was not trivial to access, since the pump frequency at which $\delta' = 0$ itself depends on P_p . To fix this, we have marked the points $n_c(\delta' = 0)$ in Supplementary Fig. 15c with a star symbol and refer to this figure in the caption of Fig. 5.

Change to manuscript:

We have modified Supplementary Fig. 15 to explicitly show the experimental n_c values for $\delta' = 0$ as star symbols and added a corresponding sentence to the caption. Furthermore, we have added a note to the caption of Fig. 5, which refers to both Fig. 4 for the color-code used for the three backaction datasets and Supplementary Fig. 15 for the pump photon numbers.

Reviewer #2:

“8. As per my understanding the authors demonstrate a fano feature in PPIT/OMIT response with a red sideband for the first time which is particularly interesting. The authors provide a detailed theoretical

analysis in supplementary note V (F, I) and, VI (B) which explains the origin of fano response. Previous optomechanical systems (e.g. ref.34) observe OMIT arising from dissipative coupling but without such fano features. What enables the authors to observe such fano PPIT response? Is it primarily due to stronger g_k or some additional contributing factor?”

Reply:

We thank the reviewer for this important and interesting question. In the sideband-resolved limit and with a pump around one of the sidebands, i.e., for $|\Delta_p| \gg \kappa_0$, the Fano interference effect for an external-dissipative coupling will be very small, because the external-dissipative coupling mainly leads to an effective enhancement of the dispersive coupling rate (very different from the internal-dissipative coupling). This explains why the OMIT in Ref. [34] does not show a Fano response. We detail this on a formal level in the new Supplementary Note VI D. For other experimental settings, e.g. pumping near HF resonance or in the bad cavity limit, the situation will be different and Fano resonances will appear depending on the exact parameters.

Change to manuscript:

We have added a statement regarding the absence of Fano in Ref. [34] to the new Supplementary Note VI D, which discusses the expected modifications of coupling rates, dynamical backaction and PPIT with non-negligible external-dissipative coupling. The specific statement can be found in the lines 567-570.

Reviewer #2:

“9. Regarding SQUID optomechanics, I suggest that the authors additionally cite the following relevant works for completeness and context: 10.1038/s41598-022-05438-x and 10.1038/s42005-020-00514-y”

Reply:

We thank the reviewer for this suggestion and included these references as well as the related paper by Zoepfl *et al.*, Phys. Rev. Lett. **130**, 033601 (2023).

Change to manuscript:

We added the new references [43], [45], and [46].

Reviewer #2:

“10. Distinction of internal vs. external decay modulation: The dissipative coupling reported here arises from modulation of internal decay rate. Can the authors highlight the results that specifically depends on κ_i modulation which would otherwise be not present with κ_{ext} modulation? Additionally, can similar effects be achieved in circuits that exhibit external dissipative coupling?”

Reply:

We are very excited to see such large interest in the external-dissipative photon-pressure by both reviewers, since we are currently working on its experimental implementation with the next device generation. We must admit though, that these specific questions are hard to answer in the most general sense for arbitrary HF, LF and coupling parameters. Therefore, we restrict our considerations to the parameter space of the experiment described in the current manuscript (in good approximation sideband-resolved, red-sideband pumping, opposite signs of $g_{0\omega}$ and $g_{0\kappa}$ etc.).

In order not to repeat the extensive new texts from the manuscript/Supplementary Material here in the reply, we kindly refer the reviewer to the relevant new text passages and discussions in the revised manuscript/SM, which are listed below.

Change to manuscript:

We have added several short sections in the main manuscript highlighting the difference between internal-dissipative and external-dissipative coupling at the appropriate locations (lines 307-311, lines 342-348, lines 445-459). Furthermore, we have added a new subsection to the Supplementary Material (Note VI D, lines 545-597), which treats a possible existence of $g_{\kappa_{\text{ext}}}$ on a formal level and discusses expected differences to our observations, also addressing at the same time the corresponding comments by reviewer #1.